# GENERATIVE TIME SERIES LEARNING WITH TIME-FREQUENCY FUSED ENERGY-BASED MODEL

## ABSTRACT

Long-term time series forecasting has gained significant attention in recent academic research. However, existing models primarily focus on deterministic point forecasts, neglecting generative long-term probabilistic forecasting and pre-training models across diverse time series analytic tasks, which are essential for uncertainty quantification and computational efficiency. In this paper, we propose a novel encoder-only generative model for long-term probabilistic forecasting and imputation. Our model is an *energy-based model*, employing a time-frequency block to construct an unnormalized probability density function over temporal paths. The time-frequency block consists of two key components, i.e., a residual dilated convolutional network to increase the receptive fields of raw time series, and a time and frequency features extracting network to integrate both local and global patterns. Our design enables the prediction of long-term time series in a single forward run using Langevin MCMC, which drastically improves the efficiency and accuracy of long-term forecasting. Moreover, our model naturally serves as a general framework for forecasting at varying prediction lengths and imputing missing data points with one pre-trained model, saving both time and resources. Experiments demonstrate that our model achieves competitive results in both forecasting and imputation tasks across a diverse range of public datasets, highlighting its potential as a promising approach for a unified time series forecasting model capable of handling multiple tasks.

## 1 INTRODUCTION

Time series forecasting and interpolation are widely used in many real-world applications, including financial analysis (Tsay, 2005), weather forecasting (Wu et al., 2021), imputation of missing data for data mining (Friedman, 1962), and anomaly detection (Yang et al., 2023). Furthermore, long-range time series forecasting is crucial in areas such as inventory planning. The practical value of time series analysis has led to great interest in the field.

A wide variety of deep learning architectures have been developed to address the diversity of time series datasets in different domains. These architectures can be broadly classified into three groups: 1) Auto-regressive models using RNNs (LSTM or GRU) (Salinas et al., 2020) are the simplest to implement, but suffer from error accumulation and limited memory capacity to capture long-range dependencies. 2) Transformer-based models (Wen et al., 2022) are the most popular methods and have achieved state-of-the-art results. However, they typically have quadratic complexity and cannot easily quantify uncertainties. 3) Structured models (deep SSMs) (Rangapuram et al., 2018; Kurle et al., 2020; Sun et al., 2022) combine classical state space models (SSM) with deep learning, but can still suffer from error accumulation.

Despite the impressive results achieved by the models mentioned above, most of them have limitations in addressing long-term forecasting coherently. In other words, they typically maximize the likelihood of each temporal step but cannot guarantee the maximal likelihood of the whole sequence. This can lead to inaccurate forecasts, especially for long-term forecasting tasks. To address this limitation, we propose a novel **T**ime-**F**requency fused **E**nergy-**B**ased **M**odel (TF-EBM), a univariate generative time series model employing energy-based method. TF-EBM is a probabilistic model that can characterize the probability distribution of the whole multi-step future values (i.e., *path*) of the time series, which is valuable for downstream decision-making business tasks depending on forecasts.

The two key ingredients of TF-EBM include *energy-based model* (EBM) (Hinton, 2002; LeCun et al., 2006; Ranzato et al., 2007; Du & Mordatch, 2019; Xie et al., 2016; 2018) and *time-frequency fusion*.

Firstly, EBMs have been successfully applied to various machine learning tasks, including image and text generation (Deng et al., 2020). Compared with well-defined distribution form such as Guassian, EBM is a non-normalized flexible high-dimensional distribution model (Song & Kingma, 2021) that only requires modeling the energy function, which benefits the design of deep models. Specifically, a probability density or mass function up to a normalizing constant is defined and then EBM adapts to the real complex joint distribution of time series' temporal paths. In principle, this approach is ideal for modeling sequences such as time series because the whole sequence can be evaluated at once, achieving long-range temporal coherence.

Secondly, the deficiency of modeling long-range dependencies poses a significant challenge in forecasting long-term time series, and our objective is to address this challenge by capturing both local and global patterns. Inspired by the WaveNet (Oord et al., 2016), we use a stack of dilated convolutional neural network (CNN) to extract local pattern within the temporal domain, as well as global patterns within the frequency domain. Our architecture can also increase the receptive field in the frequency domain in addition to the time domain. CNNs are originally designed for image data, and are adapted here to time series forecasting by applying multiple layers of causal CNN (Oord et al., 2016; Borovykh et al., 2017), i.e., masking the future information. According to the spectral transform theorem (Katznelson, 2004; Chi et al., 2019; 2020), the frequency domain contains the long-term or global information of the time series. Unlike other frequency-domain-based time series models (Zhou et al., 2022; Wu et al., 2022) that only keep some frequency modes, we apply multiple CNN layers to automatically extract the relevant frequency modes, and by further integrating both temporal and frequency features, TF-EBM can better capture the complex and long-range dependencies.

To train TF-EBM, we use maximum likelihood estimation with gradient-based Markov Chain Monte Carlo (MCMC). This approach allows us to learn parameters of the energy function such that regions of high density of training data produce lower energy than elsewhere. The sampling is done with short-run Langevin MCMC (Nijkamp et al., 2019; Pang et al., 2020; Xie et al., 2022) for better efficiency than the traditional MCMC (see Appendix D for complexity analysis).

In addition, harnessing the power of EBM-based generative model, TF-EBM can act as a "pre-trained" time series model by varying the look-back window and prediction length while fixing the total length of forecasting tasks, or by adjusting the percentage of masked data points for imputation tasks. Once trained, TF-EBM can be used to generate new time series samples, or to fill in missing values in existing time series.

In summary, our main contributions are as follows:

- Motivated by the successful application of energy model in image generation and deep temporal/frequency domain analysis of time series in recent years, we introduce TF-EBM, a novel probabilistic time series analysis model leveraging energy-based method for parameter-efficient time series learning.

- In probabilistic forecasting, long-term point forecasting and imputation tasks, TF-EBM shows competitive forecasting performance compared with other state-of-the-art probabilistic and long-term point forecasting models.

- TF-EBM is capable of handling both varying prediction lengths for the forecasting task and varying proportions of missing data for the imputation task with one single trained model, by characterizing the distribution of temporal path.

We believe that as a highly versatile generative model, TF-EBM provides a unified solution for multiple tasks of both short-term and long-term time series.

## 2  RELATED WORKS

In this section, we review several related works in the area of energy-based models, transformers, and probabilistic time series models.

## 2.1 Energy-based Time Series Models

EBM has a long history in machine learning. However, to the best of our knowledge, there have been few attempts to directly apply EBM to time series data, among which, TimeGrad (Rasul et al., 2021) is a diffusion probabilistic time series forecasting model that is also based on EBM, transforming the data distribution to the target distribution by slowly injecting noise. However, TimeGrad is an RNN conditioned diffusion probabilistic model, suffering from error accumulation and high complexity in the long-range forecasting scenario due to its auto-regressive nature. In addition, as a DDPM (Denoising Diffusion Probabilistic Models) model, it is sensitive to the scale of noise and the sampling is slow Yan et al. (2021). ScoreGrad (Yan et al., 2021) extends the TimeGrad model by replacing the discrete diffusion steps with continuous Stochastic Differential Equation (SDE), but as other auto-regressive models, it still suffers from the sampling inefficiency by sampling future target step-by-step.

TF-EBM differs from both TimeGrad and ScoreGrad, where the probability distribution across dimensions is modeled at every time point. Instead, the distribution of the whole sequence for each dimension is modeled in TF-EBM, which not only makes it more efficient, but also better keeps long-term coherence compared to auto-regressive models.

## 2.2 Non-EBM-based Time Series Models

In this section, we briefly review some non-energy-based time series models.

**Transformer-based Models**   Transformer-base models (Vaswani et al., 2017) have achieved tremendous success in capturing long-range dependencies in modeling time series (Wen et al., 2023), among which Autoformer (Wu et al., 2021), Pyraformer (Liu et al., 2021), ETSFormer (Woo et al., 2022), Quatformer (Chen et al., 2022), Non-stationary Transformer (Liu et al., 2022), and FEDformer (Zhou et al., 2022) are some recent advances. These methods typically employ an encoder-decoder attention architecture. Specifically, FEDformer applies the attention on multiple sequence frequencies obtained from Fourier transformation (frequency domain only). To further improve learning capacity for long-term patterns, our TF-EBM model extracts features of both temporal and frequency domains.

**Probabilistic Models**   There are several other probabilistic time series models, where DeepAR (Salinas et al., 2020) is an RNN based auto-regressive model with model distribution depending on the RNN output. Deep State Space (Rangapuram et al., 2018) is another structured state space model (SSM) with model distribution depending on the latent variable dynamics. ARSGLS and its variant RSGLS-ISSM model (Kurle et al., 2020) improve Deep SSM with additional regime switching variable mechanism to shift among different linear dynamics. The above probabilistic models generate multi-step future values step-by-step, which results in error accumulation, while our TF-EBM model generates multiple steps at once based on the joint distribution.

# 3 TF-EBM (Time-Frequency fused Energy-Based Model)

## 3.1 Problem Formulation

A univariate generative time series forecasting problem is defined as follows: Let $y_t \in \mathbb{R}^1$ be the time series target at time $t$. Given the conditional lookback window $T : (y_1, y_2, ..., y_T)$, our goal is to forecast the next $\tau$ steps $y_{T+1:T+\tau}$, and we consider the univariate time series $y_{1:T+\tau} \in \mathbb{R}^{1 \times (T+\tau)}$ as a temporal path. TF-EBM models the real joint distribution $p_{\text{data}}(y_{1:T+\tau})$ of path $y_{1:T+\tau}$ via $p_\theta(y_{1:T+\tau})$ parameterized with $\theta$. After training, one can sample $y_{T+1:T+\tau}$ given $y_{1:T}$ according to the distribution $p_\theta(y_{1:T+\tau})$.

## 3.2 Energy-Based Models and Sampling

Under the paradigm of energy-based model, the joint distribution of temporal path $y_{1:T+\tau}$ is modeled by the unnormalized probability density $e^{-E(\mathbf{y_{1:T+\tau}})}$ with the energy function $E(y_{1:T+\tau}) \in \mathbb{R}$. Accordingly, the path $y_{1:T+\tau}$ with the lowest energy has the maximum probability density. In our model, $E_\theta(y_{1:T+\tau})$ is an encoder-only multilayer deep neural network parameterized with $\theta$. Then

the joint density is defined by the Boltzmann distribution

$$p_\theta(y_{1:T+\tau}) = \frac{\exp(-E_\theta(y_{1:T+\tau}))}{Z(\theta)}, \tag{1}$$

where $Z(\theta) = \int \exp(-E_\theta(y_{1:T+\tau}))) \, dy$ is the partition function or normalizing constant. The learned distribution $p_\theta(y_{1:T+\tau})$ approximates $p_{\text{data}}(y_{1:T+\tau})$ of the training set $y_{1:T+\tau}$, and the training process is discussed in section 3.3 later.

However, generating high-quality samples for above defined EBM is challenging, for which we propose drawing samples using Langevin dynamics with the gradient of the energy function. Theoretically, the Langevin dynamics converge to the target distribution with infinite steps and infinitesimal step size. In order to improve sampling efficiency, we propose the application of the short-run MCMC to approximate sampling, which is encouraged by the successful application in the image generation tasks (Nijkamp et al., 2019; Pang et al., 2020; Xie et al., 2022). The short-run MCMC undergoes the following iterative sampling procedure:

$$z_0 \sim p_0(z), \quad z_{k+1} = z_k - \frac{\varepsilon^2}{2}\nabla_z E_\theta(z_k) + \varepsilon\omega_k, \quad \omega_k \sim \mathcal{N}(0,1), \tag{2}$$

where $p_0(z)$ is the fixed initial distribution, $\omega_k$ is the Gaussian noise term, and $z_k$ is the generated sample after $k$-steps of Langevin MCMC from $p_0$. The idea is to start from a random point $z_0$, and then gradually move toward a direction with higher probability using the gradient of $E_\theta$. One can show that under some regularity conditions, $\mathbf{z}_k$ is distributed as $p_\theta(z)$ when $\varepsilon \to 0$ and $K \to \infty$ as in (Welling & Teh, 2011). In our experiments, we sample $z_0$ from a Gaussian distribution. Once the energy model is trained, given the conditional sequence $y_{1:T}$, one can sample $y_{T+1:T+\tau}$ from $p_\theta(y_{1:T+\tau})$ using Langevin MCMC inference steps shown in Figure 1.

### 3.3 MAXIMUM LIKELIHOOD TRAINING

In this section, we describe how to train the model with maximum likelihood so that $p_\theta$ expressed by the energy function $E_\theta(y)$ approximates the target distribution $p_{\text{data}}(y)$, where we denote $y_{1:T+\tau}$ simply as $y$ for notational convenience. The loss objective function is the negative log likelihood of the data $\mathcal{L}_\theta = \mathbb{E}_{y \sim p_{\text{data}}}[-\log p_\theta(y)]$. After deriving the negative log likelihood (see Appendix A for the detailed derivation), the gradient of the loss objective function for EBM is the contrastive divergence objective summarized as follows:

$$\begin{aligned}
\nabla_\theta \mathcal{L}_\theta(y) &= \nabla_\theta \left[ \mathbb{E}_{y^+ \sim p_{\text{data}}} E_\theta(y^+) - \mathbb{E}_{y^- \sim p_\theta} E_\theta(y^-) \right] \\
&\approx \nabla_\theta \left[ \frac{1}{n}\sum_{i=1}^{n} E_\theta(y_i^+) - \frac{1}{\tilde{n}}\sum_{i=1}^{\tilde{n}} E_\theta(y_i^-) \right],
\end{aligned} \tag{3}$$

where $y^+$ and $y^-$ denote the positive and negative samples, respectively. $y^+$'s are sampled from the data distribution $p_{\text{data}}$ and $y^-$'s are sampled from the model $p_\theta(y)$ with the short-run MCMC described in Section 3.2. Additionally, during our experiments, we find that the output of the network $E_\theta(y)$ is not well-bounded. We thus add a regularization term to $E_\theta(y)$ to help with the convergence. As a result, the final gradient of the loss objective function is as follows:

$$\nabla_\theta \mathcal{L} \approx \nabla_\theta \left[ \frac{1}{n}\sum_{i=1}^{n} E_\theta(y_i^+) - \frac{1}{\tilde{n}}\sum_{i=1}^{\tilde{n}} E_\theta(y_i^-) \right] + \nabla_\theta \mathcal{L}_2(\theta), \tag{4}$$

where $\mathcal{L}_2$ is the L2 regularization term for the outputs of the the network $E_\theta(y)$.

### 3.4 ENCODER ARCHITECTURE

The direct application of EBM on time series is challenging due to two key properties, i.e., the presence of multi-period signal mixed with high-frequency noise, and the need to capture both local details and global long-term patterns. In order to address these challenges and to enhance the characterization of the time series, the encoder of TF-EBM adopts the residual Time-Frequency Block (TFB) to construct energy function $E_\theta$ as shown in Figure 1. Such organized residual way of TF-EBM can speed up convergence and enable training deeper models (He et al., 2016). Each

TFB layer comprises two components, i.e., a **D**ilatedConv-**S**iLU[1]-**C**onv1d (DSC) network and time-frequency (TF) domain features extracting network. The design of the DSC module is inspired by the wavenet (Oord et al., 2016) architecture to increase receptive fields of time series and filter out the high-frequency noise (as discussed in Section 4.4 later). This design choice greatly benefits the extraction of level-wise temporal and frequency patterns. Time-frequency domain extraction operation is discussed in detail in Section 3.5 later.

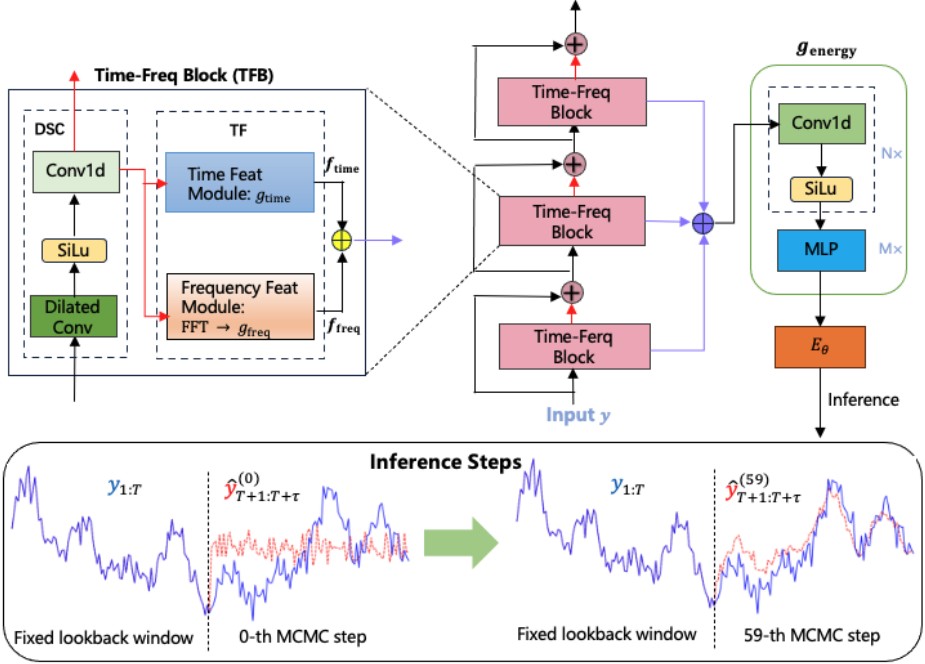

Figure 1: TF-EBM model architecture (top) with an inference example (bottom). $E_\theta$ is the ultimate scalar energy value for $y_{1:T+\tau}$. TF-EBM is constructed by the Time-Freq Block which consists of a DSC network and time-frequency feature extracting network. The extracted time and frequency features in each layer are firstly added together and then fed to the energy function block to generate the the energy $E_\theta$. In the inference steps, with the lookback window $y_{1:T}$ fixed, the inference samples $y_{T+1:T+\tau}$ (red dashed line in the 59-th MCMC step) can be sampled starting from a standard normal distribution (red dashed line in the 0-th MCMC step) by 59-step Langevin MCMC. The blue solid line denotes the true target.

Mathematically, given the lookback window $y_{1:T}$ and forecast window $y_{T+1:T+\tau}$, the overall concatenated input temporal window for the $0$-th layer is $y^0_{1:T+\tau} = [y_{1:T}, y_{T+1:T+\tau}] \in \mathbb{R}^{1 \times (T+\tau)}$, while the input to the $l$-th layer is $y^l_{1:T+\tau}$ that consists of the input to the previous ($l$-1)-th layer and the output of the DSC module:

$$y^l_{1:T+\tau} = y^{l-1}_{1:T+\tau} + \text{DSC}(y^{l-1}_{1:T+\tau}), \quad l = 1, ..., L. \tag{5}$$

### 3.5 TIME-FREQUENCY FEATURE EXTRACTION MODULE

It is well known that frequency modes of time series capture the global pattern (Chi et al., 2019; 2020; Zhou et al., 2022), while the temporal features captured in the context window are mostly local patterns, and the benefit from the integration of both global and local information will be demonstrated in the ablation study (Section 4.4 and Appendix C) on a variety of tasks. Therefore, we design temporal and frequency domain extraction modules to extract the information for the $l$-th layer skip connection $\text{DSC}(y^l_{1:T+\tau})$. Firstly, we apply the Fast Fourier Transformation (FFT) $\mathcal{F}(\text{DSC}(y^l_{1:T+\tau})) \in \mathbb{C}^{1 \times F}$ along the temporal dimension to map the time domain features to

---

[1]SiLU: Sigmoid Linear Unit activation function

frequency domain, where $F = \lfloor (T + \tau)/2 \rfloor + 1$ is the number of frequencies. Following the convention in (Chi et al., 2019; 2020), to efficiently perform global updates on spectral data, we concatenate the real and imaginary parts of FFT coefficients for each frequency mode, and then apply a learned convolutional 1D (Conv1D) layer followed by a perceptron (MLP) layer (Figure 1).

Let us define operator $g := \text{MLP}\left(\text{SiLU}(\text{Conv1d}(\cdot))\right)$. We then have the frequency domain feature $f_{\text{freq}}$ and time domain feature $f_{\text{time}}^l \in \mathbb{R}^d$ as follows:

$$
\begin{aligned}
f_{\text{freq}}^0 &= g_{\text{freq}}\left([\text{Re}(\mathcal{F}(y_{1:T+\tau}^0)),\ \text{Im}(\mathcal{F}(y_{1:T+\tau}^0))]\right), \quad l = 0, \\
f_{\text{freq}}^l &= g_{\text{freq}}([\text{Re}(\mathcal{F}(\text{DSC}(y_{1:T+\tau}^l))),\ \text{Im}(\mathcal{F}(\text{DSC}(y_{1:T+\tau}^l)))]), \quad 0 < l \le L, \\
f_{\text{time}}^0 &= g_{\text{time}}(y_{1:T+\tau}^0)), \quad l = 0, \\
f_{\text{time}}^l &= g_{\text{time}}(\text{DSC}(y_{1:T+\tau}^l))), \quad 0 < l \le L.
\end{aligned}
\tag{6}
$$

The final energy output consists of two parts, i.e., local features (multilayer time domain) and global features (multilayer frequency domain):

$$
E_\theta(y_{1:T+\tau}) = g_{\text{energy}}\left(\sum_{j=0}^{L} f_{\text{time}}^j + \sum_{j=0}^{L} f_{\text{freq}}^j\right).
\tag{7}
$$

## 4 EXPERIMENTS

**Datasets and Tasks**   To evaluate the performance of TF-EBM, we examine the forecasting and imputation tasks on eight popular public distastes (electricity, traffic, exchange rate, wiki, ETT1, etc. ) that are extensively used for benchmarking (see Table 1 for details).

Since our model is a probabilistic model, we compare with other competitive probabilistic models in terms of Continuously Ranked Probability Score (CRPS) (Matheson & Winkler, 1976). Additionally, we also compare against other transformer-based non-probabilistic models, since they often achieve the state-of-the-art (SOTA) performances on these public datasets on the long-term forecasting task in terms of metrics of mean absolute error (MAE) and mean squared error (MSE). Due to space constraints, we have included the transfer learning experiments in Appendix F .

Table 1: Summary of datasets for the experiments.

| Tasks | Datasets | Metrics | Series Length |
|---|---|---|---|
| Probabilistic Forecasting | exchange rate, electricity, traffic, wiki | CRPS | Hour: $24 \sim 168$ 
 Day: $30 \sim 150$ |
| Long-term Forecasting | ETTh1, ETTh2, ETTm1, ETTm2, exchange rate, electricity, traffic | MAE, MSE | $96 \sim 720$ |
| Imputation | ETTh1, ETTh2, ETTm1, ETTm2, electricity | MAE, MSE | 96 |

**Baselines**   To have a thorough study on the forecasting and imputation tasks, we compare against a wide range of well-known probabilistic and non-probabilistic models, including DeepAR (Salinas et al., 2020), Deep State Space (Rangapuram et al., 2018), ARSGLS and its variant RSGLS-ISSM model (Kurle et al., 2020), TimeGrad (Rasul et al., 2021) for probabilistic forecasting; Dlinear (Zeng et al., 2023), Timesnet (Wu et al., 2022), ETSformer (Woo et al., 2022), FEDformer (Zhou et al., 2022), Non-Stationary Tranformer (Liu et al., 2022), Autoformer (Wu et al., 2021), and LSTM (Hochreiter & Schmidhuber, 1997) for both long-term forecasting and imputation task.

### 4.1 FORECASTING

Forecasting capability is one of the fundamental evaluation criteria for time series models. As our model is a probabilistic model, we perform two forecasting tasks to fully evaluate our model, i.e., probabilistic forecasting and long-term forecasting on public datasets.

**Probabilistic Forecasting**   For probabilistic forecasting task, we use CRPS as the proper scoring metric, defined as

$$
\text{CRPS}(F, x) = \int_{\mathbb{R}} (F(u) - \mathbb{I}_{y \le u})^2 du,
\tag{8}
$$

where $\mathbb{I}_{y\leq u}$ is the indicator function which is 1 if $y \leq u$ and 0 otherwise. The model's cumulative distribution function (CDF) can be approximated by the empirical CDF with $n$ samples $\hat{y}_i$: $\hat{F}(u) = 1/n \sum_{i=1}^{n} \mathbb{I}\{\hat{y}_i \leq u\}$. CRPS measures the compatibility of model's CDF $F$ of the observation $y$ in comparison to other probabilistic models on both hourly and daily observed datasets listed in 1. A lower CRPS metric indicates a better probabilistic forecasting model. We evaluate in a rolling fashion (prediction length=24H or 30D) and with a single long-term (prediction length is 168H or 150D) forecast as done in Kurle et al. (2020). The results are summarized in Table 2 with additional results for the longer term (336 and 720) forecasts in Table 8 in Appendix E. Our model is on par with or surpasses other baseline models. Only ARSGLS and RSGLS-ISSM perform better than ours on the Wikipedia and exchange rate datasets, respectively. However, both models can be computationally expensive to train and forecast.

Table 2: CRPS metric (lower is better) results. Mean and std. deviation for TF-EBM and TimeGrad are computed over 5 and 3 independent runs, respectively. Other model results are taken from Kurle et al. (2020). The best results are highlighted in red.

| Models | exchange rate | | electricity | | traffic | | wiki | |
|---|---|---|---|---|---|---|---|---|
| | 30 (rolling) | 150 | 24 (rolling) | 168 | 24 (rolling) | 168 | 30 (rolling) | 150 |
| DeepAR | 0.009±0.001 | 0.019±0.002 | 0.057±0.003 | 0.062±0.004 | 0.120±0.003 | 0.138±0.001 | 0.281±0.008 | 0.855±0.552 |
| Deep State | 0.010±0.001 | 0.017±0.002 | 0.071±0.000 | 0.088±0.007 | 0.131±0.002 | 0.131±0.005 | 0.296±0.007 | 0.338±0.017 |
| RSGLS-ISSM | 0.007±0.000 | 0.014±0.001 | 0.070±0.001 | 0.091±0.004 | 0.148±0.005 | 0.206±0.002 | 0.248±0.006 | 0.345±0.010 |
| ARSGLS | 0.009±0.000 | 0.022±0.001 | 0.138±0.003 | 0.154±0.005 | 0.136±0.003 | 0.175±0.008 | 0.217±0.010 | 0.283±0.006 |
| TimeGrad | 0.009±0.000 | 0.020±0.003 | 0.054±0.003 | 0.061±0.005 | 0.112±0.004 | 0.162±0.030 | 0.268±0.005 | 0.414±0.023 |
| **TF-EBM (ours)** | 0.010±0.001 | 0.019±0.002 | 0.053±0.002 | 0.060±0.005 | 0.105±0.003 | 0.124±0.003 | 0.248±0.018 | 0.321±0.020 |

**Long-Term Forecasting** For long-term forecasting task, we use the mainstream long-term forecasting dataset, including four ETT datasets, electricity, exchange rate, and traffic. We use MAE and MSE metrics that are widely adopted, as in Timesnet (Wu et al., 2022) and Autoformer (Wu et al., 2021). And for a fair comparison, all models applied a unified experimental setup with prediction lengths of $\{96, 192, 336, 720\}$ and lookback window length of 96. We collected the baseline results from Wu et al. (2022) for other models. The complete results are shown in Table 3, where our model achieves 67.9% SOTA performance on MSE metric and 64.2% SOTA performance on MAE metric compared with other models, which indicates competitive long-term forecasting performance of TF-EBM.

## 4.2 IMPUTATION

Real-world time series data often contain missing values due to sensor malfunctions or other factors. This can degrade the performance of downstream tasks such as classification or regression. Therefore, imputation is widely used in practical applications to recover the missing values. As TF-EBM models the joint probability density of $y_{1:T+\tau}$, the temporal pattern of the missing data can be naturally recovered using the partially observed data.

We evaluate the imputation capability of our model on two public datasets, including electricity and four ETT datasets from Zhou et al. (2021). We randomly mask the data points along the time dimension at different rates (12.5%, 25%, 37.5%, and 50%) with a fixed sequence length of 96 and then recover the missing data with the corresponding mask rate. The evaluation is done in three ways: 1) infer the data at each missing proportion with TF-EBM trained at corresponding mask rate of data ("ours" in Table 4); 2) infer the data with varying missing proportions with TF-EBM trained by masking 50% of data ("TF-EBM (50%)"); 3) train a single TF-EBM with different mask ratios ("pre-trained"). As shown in Table 4, our model trained in the above three ways outperforms other baselines on the MAE/MSE metrics, which indicates that TF-EBM has superior imputation capacity and can generally recover data with varying missing proportions with one trained model.

## 4.3 PRE-TRAINING MODEL ACROSS DIFFERENT TASKS

Pre-training has yielded remarkable outcomes in the field of Natural Language Processing (NLP), with models like Bert (Devlin et al., 2018) and GPT (Radford et al., 2018) achieving state-of-the-art results on a variety of tasks. One of the significant advantages of pre-training is that it decouples

Table 3: MSE/MAE (lower is better) results for different prediction lengths and the same lookback window length 96. Our model results are computed over 5 independent runs. Results of all other methods are taken from Wu et al. (2022). The best result is highlighted in red.

| models | | TF-EBM (ours) | | TimesNet | | ETSformer | | Dlinear | | FEDformer | | Stationary | | Autoformer | | LSTM | |
|---|---|---|---|---|---|---|---|---|---|---|---|---|---|---|---|---|---|
| metric | | MSE | MAE | MSE | MAE | MSE | MAE | MSE | MAE | MSE | MAE | MSE | MAE | MSE | MAE | MSE | MAE |
| ETTh1 | 96 | 0.382 | 0.400 | 0.384 | 0.402 | 0.494 | 0.479 | 0.386 | 0.400 | 0.376 | 0.419 | 0.513 | 0.491 | 0.449 | 0.459 | 1.044 | 0.773 |
| | 192 | 0.437 | 0.433 | 0.436 | 0.429 | 0.538 | 0.504 | 0.437 | 0.432 | 0.420 | 0.448 | 0.534 | 0.504 | 0.500 | 0.482 | 1.217 | 0.832 |
| | 336 | 0.472 | 0.450 | 0.491 | 0.469 | 0.574 | 0.521 | 0.481 | 0.459 | 0.459 | 0.465 | 0.588 | 0.535 | 0.521 | 0.496 | 1.259 | 0.841 |
| | 720 | 0.479 | 0.477 | 0.521 | 0.500 | 0.562 | 0.535 | 0.519 | 0.516 | 0.506 | 0.507 | 0.643 | 0.616 | 0.514 | 0.512 | 1.271 | 0.838 |
| ETTh2 | 96 | 0.297 | 0.349 | 0.340 | 0.374 | 0.340 | 0.391 | 0.333 | 0.387 | 0.358 | 0.397 | 0.476 | 0.458 | 0.346 | 0.388 | 2.522 | 1.278 |
| | 192 | 0.368 | 0.400 | 0.402 | 0.414 | 0.430 | 0.439 | 0.477 | 0.476 | 0.429 | 0.439 | 0.512 | 0.493 | 0.456 | 0.452 | 3.312 | 1.384 |
| | 336 | 0.411 | 0.433 | 0.452 | 0.452 | 0.485 | 0.479 | 0.594 | 0.541 | 0.496 | 0.487 | 0.552 | 0.551 | 0.482 | 0.486 | 3.291 | 1.388 |
| | 720 | 0.428 | 0.453 | 0.462 | 0.468 | 0.500 | 0.497 | 0.831 | 0.657 | 0.463 | 0.474 | 0.562 | 0.560 | 0.515 | 0.511 | 3.257 | 1.357 |
| ETTm1 | 96 | 0.339 | 0.374 | 0.338 | 0.375 | 0.375 | 0.398 | 0.345 | 0.372 | 0.379 | 0.419 | 0.386 | 0.398 | 0.505 | 0.475 | 0.863 | 0.664 |
| | 192 | 0.377 | 0.396 | 0.374 | 0.387 | 0.408 | 0.410 | 0.380 | 0.389 | 0.426 | 0.441 | 0.459 | 0.444 | 0.553 | 0.496 | 1.113 | 0.776 |
| | 336 | 0.432 | 0.432 | 0.410 | 0.411 | 0.435 | 0.428 | 0.413 | 0.413 | 0.445 | 0.459 | 0.495 | 0.464 | 0.621 | 0.537 | 1.267 | 0.832 |
| | 720 | 0.470 | 0.456 | 0.478 | 0.450 | 0.499 | 0.462 | 0.474 | 0.453 | 0.543 | 0.490 | 0.585 | 0.516 | 0.671 | 0.561 | 1.324 | 0.858 |
| ETTm2 | 96 | 0.176 | 0.266 | 0.187 | 0.267 | 0.189 | 0.280 | 0.193 | 0.292 | 0.203 | 0.287 | 0.192 | 0.274 | 0.255 | 0.339 | 2.041 | 1.073 |
| | 192 | 0.237 | 0.301 | 0.249 | 0.309 | 0.253 | 0.319 | 0.284 | 0.362 | 0.269 | 0.328 | 0.280 | 0.339 | 0.281 | 0.340 | 2.249 | 1.112 |
| | 336 | 0.302 | 0.342 | 0.321 | 0.351 | 0.314 | 0.357 | 0.369 | 0.427 | 0.325 | 0.366 | 0.334 | 0.361 | 0.339 | 0.372 | 2.568 | 1.238 |
| | 720 | 0.398 | 0.403 | 0.408 | 0.403 | 0.414 | 0.413 | 0.554 | 0.522 | 0.421 | 0.415 | 0.417 | 0.413 | 0.433 | 0.432 | 2.720 | 1.287 |
| exchange rate | 96 | 0.085 | 0.205 | 0.107 | 0.234 | 0.085 | 0.204 | 0.088 | 0.218 | 0.148 | 0.278 | 0.111 | 0.237 | 0.197 | 0.323 | 1.453 | 1.049 |
| | 192 | 0.171 | 0.296 | 0.226 | 0.344 | 0.182 | 0.303 | 0.176 | 0.315 | 0.271 | 0.380 | 0.219 | 0.335 | 0.300 | 0.369 | 1.846 | 1.179 |
| | 336 | 0.287 | 0.391 | 0.367 | 0.448 | 0.348 | 0.428 | 0.313 | 0.427 | 0.460 | 0.500 | 0.421 | 0.476 | 0.509 | 0.524 | 2.136 | 1.231 |
| | 720 | 0.843 | 0.699 | 0.964 | 0.746 | 1.025 | 0.774 | 0.839 | 0.695 | 1.195 | 0.841 | 1.092 | 0.769 | 1.447 | 0.941 | 2.984 | 1.427 |
| electricity | 96 | 0.173 | 0.265 | 0.168 | 0.272 | 0.187 | 0.304 | 0.197 | 0.282 | 0.193 | 0.308 | 0.169 | 0.273 | 0.201 | 0.317 | 0.375 | 0.437 |
| | 192 | 0.186 | 0.278 | 0.184 | 0.289 | 0.199 | 0.315 | 0.196 | 0.285 | 0.201 | 0.315 | 0.182 | 0.286 | 0.222 | 0.334 | 0.442 | 0.473 |
| | 336 | 0.198 | 0.289 | 0.198 | 0.300 | 0.212 | 0.329 | 0.209 | 0.301 | 0.214 | 0.329 | 0.200 | 0.304 | 0.231 | 0.338 | 0.439 | 0.473 |
| | 720 | 0.235 | 0.319 | 0.220 | 0.320 | 0.233 | 0.345 | 0.245 | 0.333 | 0.246 | 0.355 | 0.222 | 0.321 | 0.254 | 0.361 | 0.980 | 0.814 |
| traffic | 96 | 0.542 | 0.342 | 0.593 | 0.321 | 0.607 | 0.392 | 0.650 | 0.396 | 0.587 | 0.366 | 0.612 | 0.338 | 0.613 | 0.388 | 0.843 | 0.453 |
| | 192 | 0.539 | 0.353 | 0.617 | 0.336 | 0.621 | 0.399 | 0.598 | 0.370 | 0.604 | 0.373 | 0.613 | 0.340 | 0.616 | 0.382 | 0.847 | 0.453 |
| | 336 | 0.529 | 0.335 | 0.629 | 0.336 | 0.622 | 0.396 | 0.605 | 0.373 | 0.621 | 0.383 | 0.618 | 0.328 | 0.622 | 0.337 | 0.853 | 0.455 |
| | 720 | 0.534 | 0.346 | 0.640 | 0.350 | 0.632 | 0.396 | 0.645 | 0.394 | 0.626 | 0.382 | 0.653 | 0.355 | 0.660 | 0.408 | 1.500 | 0.805 |

Table 4: Imputation task results. We randomly mask 12.5%, 25%, 37.5%, 50% time points in length-96 time series. Our model results are averaged among 5 runs with different mask ratios. "ours" means we infer the data at each missing proportion with TF-EBM trained at corresponding mask rate of data. TF-EBM (50%) means we infer the data with varying missing proportions with TF-EBM trained by masking 50% of data. "pre-trained" means we train a model across different tasks. All the other model results are taken from Wu et al. (2022). The best and the second-best results are highlighted in red and blue, respectively.

| Models | TF-EBM | | | | | | TimesNet | | ETSformer | | DLinear | | FEDformer | | Stationary | | Autoformer | |
|---|---|---|---|---|---|---|---|---|---|---|---|---|---|---|---|---|---|---|
| | ours | | 50% | | pre-trained | | | | | | | | | | | | | |
| Metric | MSE | MAE | MSE | MAE | MSE | MAE | MSE | MAE | MSE | MAE | MSE | MAE | MSE | MAE | MSE | MAE | MSE | MAE |
| ETTh1 | 0.037 | 0.072 | 0.036 | 0.070 | 0.068 | 0.097 | 0.078 | 0.187 | 0.202 | 0.329 | 0.201 | 0.306 | 0.117 | 0.246 | 0.094 | 0.201 | 0.103 | 0.214 |
| ETTh2 | 0.020 | 0.051 | 0.020 | 0.051 | 0.032 | 0.069 | 0.049 | 0.146 | 0.367 | 0.436 | 0.142 | 0.259 | 0.163 | 0.279 | 0.053 | 0.152 | 0.055 | 0.156 |
| ETTm1 | 0.015 | 0.044 | 0.014 | 0.043 | 0.015 | 0.045 | 0.027 | 0.107 | 0.120 | 0.253 | 0.093 | 0.206 | 0.062 | 0.177 | 0.036 | 0.126 | 0.051 | 0.150 |
| ETTm2 | 0.010 | 0.033 | 0.010 | 0.033 | 0.016 | 0.045 | 0.022 | 0.088 | 0.208 | 0.327 | 0.096 | 0.208 | 0.101 | 0.215 | 0.026 | 0.099 | 0.029 | 0.105 |
| electricity | 0.064 | 0.167 | 0.034 | 0.061 | 0.024 | 0.059 | 0.092 | 0.210 | 0.214 | 0.339 | 0.132 | 0.260 | 0.130 | 0.259 | 0.100 | 0.218 | 0.101 | 0.225 |

the base model from the downstream tasks. Consequently, we do not need to train a specific model for each downstream task from scratch because we can simply fine-tune the base model using task-specific data. Inspired by the success of pre-training in NLP and TF-EBM's ability to model probability density $p_\theta(y_{1:T+\tau})$ of an entire temporal path $y_{1:T+\tau}$, we propose a new pre-training approach for time series analysis using TF-EBM. Our pre-training approach consists of two tasks: next-step prediction and imputation. Specifically, we pre-train the base TF-EBM model by varying prediction length and masking different percentages of input series. These two tasks bear resemblance to the next sentence prediction and masked language model tasks utilized in NLP pre-training. Once a TF-EBM model is pre-trained, we can fine-tune the model for specific forecasting or imputation tasks with only two epochs. The forecasting and imputation results presented in Table 5 and 4 demonstrate that pre-trained TF-EBM performs comparably to the one conventionally trained for each dataset.

## 4.4 ABLATION STUDY

**Residual Dilated Convolutional Network**  In order to verify the impact of noise and signals in time series on the residual dilated convolutional network (= DSC network + skip connection), we perform an experiment using synthetic data. The artificial data is generated by superposition of data from five different periods and Gaussian noise as shown as the orange line in Figure 2 in Appendix C.1. For the ablation study, we conduct the forecasting experiment separately using 1-layer and 3-layer TF-EBM models with results shown in the top two panels of Figure 2. The results clearly demonstrate that TF-EBM with more layers effectively captures the underlying signal of the raw artificial data. Additionally, we analyzed the intensity of the spectrum in different layers of the residual dilated convolutional network, depicted in the bottom plot of Figure 2. This analysis reveals that the high-frequency noise diminishes in deeper layers of the model. Consequently, the ablation study strongly suggests that utilizing a stack of the residual dilated convolutional network can filter out the high-frequency noise, allowing the model to focus on capturing the desired signals in the data.

**Time and Frequency Feature Extraction**  We also conduct the ablation study experiments for the contributions of the time domain(TD) and frequency domain(FD) extraction module of our model as shown in Table 6 in Appendix C.2. Compared with the model equipped with only TD or FD, the model equipped with time-frequency feature extracting module performs best on different datasets and forecasting ranges. Therefore, both TD and FD are important factors for better model forecasting performance, which indicates that they both contribute to better modeling the energy of the data distribution. In addition, we show in Figure 3 in the Appendix the predicted mean with only TD, FD, and both TD and FD of one hourly time series from the electricity dataset. The blue line with both TD and FD best predicts the ground truth (red line). The ablation study results are consistent with the intuition that the distribution of time series can be measured from two perspectives: time domain and frequency domain, and the utilization of both domains contributes to better forecasting performance.

## 4.5 CONCLUSION AND FUTURE WORK

This paper presented a novel generative energy-based model (TF-EBM) for time series forecasting and imputation, employing multilayer time-frequency block equipped with residual dilated network and time-frequency domain extraction module. As a result, TF-EBM is able to filter high-frequency noise and extract both the local and global patterns of time series. Experiments showed that TF-EBM achieved competitive forecasting performance compared to other probabilistic and long-term point forecasting models. Additionally, TF-EBM can be pre-trained to handle varying prediction and imputation tasks with a single forecasting model, similar to the pre-trained paradigm used for NLP. It is therefore a promising approach using one single unified forecasting model for multiple tasks on time series. For the limitations and future research of our model: 1) Our model is a global univariate model without explicitly considering the dependencies among time series. Extending TF-EBM to a multivariate model so that we can simultaneously model the joint distribution among series and along time steps can further benefit the multivariate forecasting performance. Furthermore, our model can then be applied to classification and anomaly detection tasks. 2) The generalization of the energy-based architecture for time series with different backbones such as the transformer is left for further exploration. 3) Investigate alternative time and frequency feature fusion methods.

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

## A  DERIVATION OF THE ENERGY MODEL

$$
\begin{aligned}
\nabla_\theta \mathbb{E}_{y \sim p_{\text{data}}} \log p_\theta(y) &= \nabla_\theta \mathbb{E}_{y \sim p_{\text{data}}} \log p_\theta(y) \\
&= \nabla_\theta \mathbb{E}_{y \sim p_{\text{data}}} (-E_\theta(y) - \log Z(\theta)) \\
&= \nabla_\theta (-E_{y \sim p_{\text{data}}} E_\theta(y) - \log Z(\theta)) \\
&= -\nabla_\theta \mathbb{E}_{y \sim p_{\text{data}}} E_\theta(y) - \nabla_\theta \log Z(\theta) \\
&= -\nabla_\theta \mathbb{E}_{y \sim p_{\text{data}}} E_\theta(y) - \frac{\nabla_\theta Z(\theta)}{Z(\theta)} \\
&= -\nabla_\theta \mathbb{E}_{y \sim p_{\text{data}}} E_\theta(y) + \frac{\int e^{-E_\theta(y)} \nabla_\theta E_\theta(y) dy}{Z(\theta)} \\
&= -\nabla_\theta \mathbb{E}_{y \sim p_{\text{data}}} E_\theta(y) + \int \underbrace{\frac{e^{-E_\theta(y)}}{Z(\theta)}}_{p_\theta(y)} \nabla_\theta E_\theta(y) dy \\
&= -\nabla_\theta \mathbb{E}_{y \sim p_{\text{data}}} E_\theta(y) + \mathbb{E}_{y \sim p_\theta(y)} \nabla_\theta E_\theta(y) \\
&\approx \nabla_\theta \left[ \frac{1}{\tilde{n}} \sum_{i=1}^{\tilde{n}} E_\theta(y_i^-) - \frac{1}{n} \sum_{i=1}^{n} E_\theta(y_i^+) \right]
\end{aligned}
\tag{9}
$$

## B  PRE-TRAINING AND NON-PRE-TRAINING RESULTS

This section showcases TF-EBM's pre-training capabilities, as summarized in Table 5.

Table 5: MSE and MAE results (lower is better) for pre-trained and non-pretrained models on five datasets with varying prediction lengths, with the best MSE and MAE results highlighted in red. Mean and standard deviation are obtained with three independent runs. The pre-trained model achieves comparable results with models individually trained for each dataset, regardless of prediction length.

| Dataset | Pred Length | Pre-trained | | Non Pre-trained | |
|---|---|---|---|---|---|
| | | MSE | MAE | MSE | MAE |
| ETTh1 | 12 | 0.289±0.003 | 0.348±0.002 | 0.326±0.048 | 0.375±0.040 |
| | 24 | 0.320±0.007 | 0.368±0.008 | 0.335±0.020 | 0.380±0.019 |
| | 36 | 0.348±0.004 | 0.383±0.004 | 0.378±0.036 | 0.407±0.031 |
| | 48 | 0.374±0.002 | 0.399±0.004 | 0.394±0.026 | 0.420±0.024 |
| ETTh2 | 12 | 0.147±0.003 | 0.253±0.005 | 0.148±0.016 | 0.252±0.018 |
| | 24 | 0.179±0.001 | 0.277±0.003 | 0.190±0.019 | 0.284±0.015 |
| | 36 | 0.231±0.022 | 0.325±0.032 | 0.232±0.039 | 0.322±0.044 |
| | 48 | 0.246±0.015 | 0.325±0.017 | 0.259±0.011 | 0.334±0.007 |
| ETTm1 | 12 | 0.181±0.007 | 0.268±0.008 | 0.186±0.022 | 0.276±0.021 |
| | 24 | 0.292±0.009 | 0.343±0.005 | 0.277±0.005 | 0.333±0.005 |
| | 36 | 0.430±0.002 | 0.408±0.003 | 0.436±0.012 | 0.421±0.003 |
| | 48 | 0.538±0.006 | 0.451±0.002 | 0.537±0.030 | 0.452±0.012 |
| ETTm2 | 12 | 0.086±0.002 | 0.189±0.006 | 0.083±0.004 | 0.181±0.006 |
| | 24 | 0.113±0.000 | 0.216±0.000 | 0.114±0.000 | 0.217±0.001 |
| | 36 | 0.140±0.003 | 0.246±0.006 | 0.138±0.000 | 0.239±0.000 |
| | 48 | 0.168±0.004 | 0.273±0.006 | 0.185±0.018 | 0.296±0.028 |
| electricity | 12 | 0.126±0.001 | 0.228±0.001 | 0.121±0.002 | 0.222±0.003 |
| | 24 | 0.144±0.001 | 0.239±0.001 | 0.139±0.003 | 0.236±0.004 |
| | 36 | 0.167±0.001 | 0.258±0.002 | 0.167±0.014 | 0.260±0.015 |
| | 48 | 0.199±0.006 | 0.281±0.005 | 0.195±0.003 | 0.277±0.002 |

## C  ABLATION STUDY RESULTS

### C.1  RESIDUAL DILATED CONVOLUTIONAL NETWORK

In this section, we illustrate the ability of Residual Dilated Convolutional Network to automatically extract meaningful signals instead of noise using synthetic data in Figure 2.

### C.2  TIME AND FREQUENCY FEATURE EXTRACTION

In this section, we illustrate the contribution of the time and feature extraction module in Figure 3 and Table 6.

Table 6 clearly demonstrates that TF-EBM consistently outperforms other models across various datasets and forecast horizons when employing both time and frequency feature extraction modules. In addition to this, we have also incorporated additional metrics such as the average lag-1 auto-correlation, standard deviation, and the four dominant power spectra for each dataset. Upon observation, it becomes evident that TF-EBM consistently achieves superior performance on the majority of datasets and forecast horizons. There are only three exceptions, namely the ETTm1 and traffic datasets, where either the time or frequency features individually produced slightly better results. However, it is important to note that the margin of improvement in these cases is minimal and lacks statistical significance. Therefore, we firmly believe that incorporating both time and frequency features is crucial for capturing both local and global patterns within the time series.

### C.3  LONG-TERM DEPENDENCE

In this section, we performed ablation experiments on synthetic data to investigate the impact of long-term dependence. Once the model was trained, we conducted future time step forecasting using one, two, and three layers of the Time-Freq Block individually. To better visualize the results, we applied FFT to the different forecasts, as depicted in Figure 4. The figure clearly demonstrates that the forecast generated by the three-layer Time-Freq Block effectively captures the low-frequency component, indicating that the stacked Time-Freq Blocks play a significant role in capturing long-term patterns.

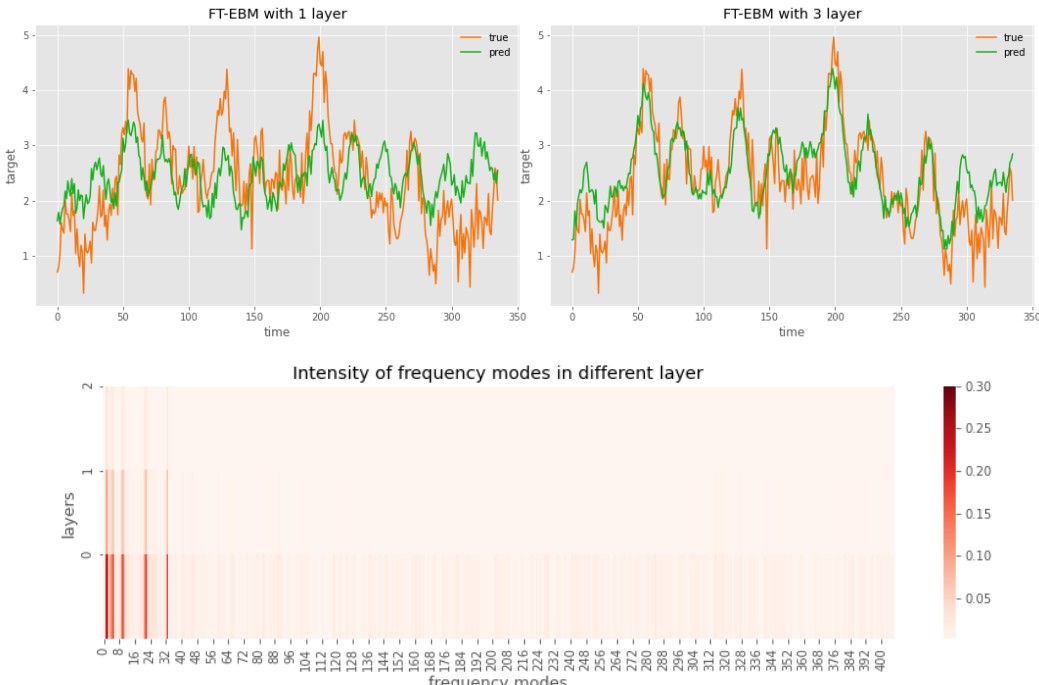

Figure 2: Ablation study of the synthetic data. Prediction plot of TF-EBM with 1 layer and 3 layers (orange line in the top two panels denotes ground truth, and green line denotes the prediction) and visualization of the intensity of spectrum in TF-EBM with 3 layers (bottom panel). The artificial data (orange line) is generated by superposition of data from five different periods and Gaussian noise. TF-EBM with 3 layers best tracks the signal of time series and high-frequency noise diminishes in the deeper layer.

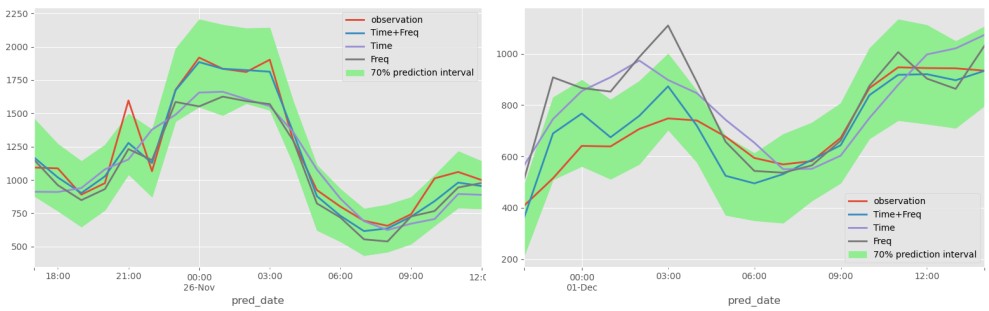

Figure 3: TF-EBM prediction intervals with time feature, frequency feature, and both time and frequency features for one series on two different days from the electricity dataset. The ground truth is in red. TF-EBM with both temporal and frequency features best tracks the ground truth.

Table 6: MSE and MAE (lower is better) results for the ablation study of using either time or frequency feature. The best result is highlighted in red. We also included the average of lag-1 auto-correlation coefficient, standard deviation, and four dominant power spectrum for each dataset.

| Dataset | Auto Corr. & Std.Dev. | Power Spectrum (1, 2, 3, 4) | Pred. Length | Time+Freq MSE | Time+Freq MAE | Time MSE | Time MAE | Freq MSE | Freq MAE |
|---|---|---|---|---|---|---|---|---|---|
| ETTh1 | 0.918 | 21629.6 | 96 | 0.382 | 0.400 | 0.591 | 0.524 | 0.404 | 0.415 |
|  | 3.252 | 10036.7 | 192 | 0.437 | 0.433 | 0.716 | 0.570 | 1.13 | 0.67 |
|  |  | 9290.1 | 336 | 0.472 | 0.450 | 0.724 | 0.582 | 0.93 | 0.65 |
|  |  | 8525.3 | 720 | 0.479 | 0.477 | 0.559 | 0.529 | 0.483 | 0.483 |
| ETTh2 | 0.967 | 46465.0 | 96 | 0.297 | 0.349 | 0.350 | 0.401 | 0.318 | 0.371 |
|  | 7.637 | 33510.7 | 192 | 0.368 | 0.400 | 0.431 | 0.440 | 1.029 | 0.672 |
|  |  | 25504.5 | 336 | 0.411 | 0.433 | 0.463 | 0.470 | 0.721 | 0.584 |
|  |  | 20932.6 | 720 | 0.428 | 0.453 | 0.487 | 0.496 | 1.595 | 0.885 |
| ETTm1 | 0.971 | 86596.0 | 96 | 0.339 | 0.374 | 0.359 | 0.390 | 0.625 | 0.514 |
|  | 3.256 | 40039.2 | 192 | 0.377 | 0.396 | 0.712 | 0.565 | 0.744 | 0.563 |
|  |  | 37215.8 | 336 | 0.432 | 0.432 | 0.426 | 0.429 | 2.814 | 0.966 |
|  |  | 34099.4 | 720 | 0.470 | 0.456 | 0.764 | 0.592 | 0.732 | 0.573 |
| ETTm2 | 0.987 | 186089.0 | 96 | 0.176 | 0.266 | 0.219 | 0.309 | 0.445 | 0.428 |
|  | 7.646 | 134118.5 | 192 | 0.237 | 0.301 | 0.287 | 0.347 | 0.686 | 0.504 |
|  |  | 102153.0 | 336 | 0.302 | 0.342 | 0.310 | 0.352 | 1.383 | 0.769 |
|  |  | 83916.0 | 720 | 0.398 | 0.403 | 0.451 | 0.445 | 1.704 | 0.891 |
| electricity | 0.907 | 16016495.3 | 96 | 0.173 | 0.265 | 0.238 | 0.287 | 0.193 | 0.277 |
|  | 1141.6 | 7805853.1 | 192 | 0.186 | 0.278 | 0.730 | 0.640 | 0.441 | 0.449 |
|  |  | 4889366.2 | 336 | 0.198 | 0.289 | 0.205 | 0.298 | 0.644 | 0.521 |
|  |  | 3984581.6 | 720 | 0.235 | 0.319 | 0.258 | 0.340 | 0.242 | 0.325 |
| exchange | 0.999 | 394.0 | 96 | 0.085 | 0.205 | 0.183 | 0.311 | 0.218 | 0.330 |
|  | 0.099 | 172.1 | 192 | 0.171 | 0.296 | 0.261 | 0.382 | 0.266 | 0.371 |
|  |  | 146.8 | 336 | 0.287 | 0.391 | 0.866 | 0.710 | 0.297 | 0.396 |
|  |  | 120.2 | 720 | 0.843 | 0.699 | 2.079 | 1.144 | 1.375 | 0.893 |
| traffic | 0.840 | 364.0 | 96 | 0.542 | 0.342 | 0.510 | 0.337 | 0.522 | 0.358 |
|  | 0.046 | 160.6 | 192 | 0.539 | 0.353 | 0.559 | 0.380 | 0.535 | 0.363 |
|  |  | 110.2 | 336 | 0.529 | 0.335 | 0.561 | 0.390 | 0.571 | 0.351 |
|  |  | 82.3 | 720 | 0.534 | 0.346 | 0.583 | 0.392 | 0.592 | 0.397 |

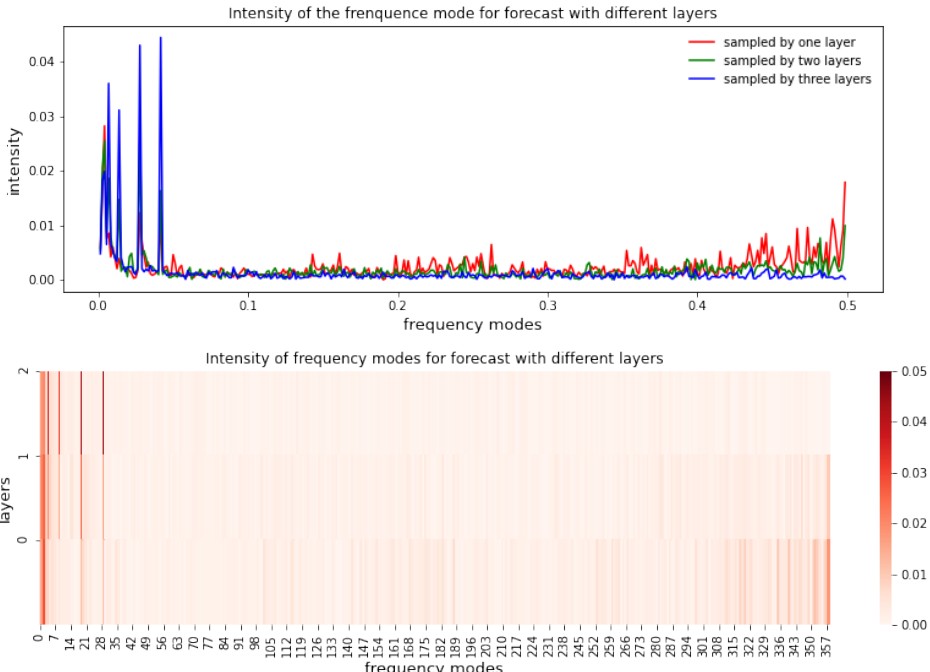

Figure 4: Ablation study of long-term dependence. The top panel shows the intensity of spectrum plot of forecast with one layer, two layers and three layers. The bottom panel illustrates the intensity of spectrum with one layer, two layers and three layers.

## D    COMPUTATIONAL COMPLEXITY ANALYSIS

In this section, we compare the theoretical complexity per layer and the total number of model parameters of TF-EBM with several benchmark models, as summarized in Table 7. We observe that TF-EBM has similar complexity to other models in both the encoder and decoder layers. Moreover, TF-EBM has significantly fewer model parameters than all other models except DLinear. Despite its smaller size, TF-EBM outperforms DLinear on almost all benchmark tests.

Table 7: Computational complexity and model size comparison of TF-EBM and benchmark models. $T$ is the length of the past series, $\tau$ is the forecast length, $C$ is the number of Langevin MCMC steps in TF-EBM, $M$ and $K$ stand for million and thousand, respectively.

| Model | Encoder layer | Decoder layer | Total number of parameters |
|---|---|---|---|
| DLinear | $\mathcal{O}(T)$ | $\mathcal{O}(T + \tau)$ | 140K |
| Autoformer | $\mathcal{O}(T \log T)$ | $\mathcal{O}((T/2 + \tau) \log(T/2 + \tau))$ | 10M |
| Pyraformer | $\mathcal{O}(T)$ | $\mathcal{O}(\tau(T + \tau))$ | 14M |
| Fedformer | $\mathcal{O}(T)$ | $\mathcal{O}(T/2 + \tau)$ | 16M |
| **TF-EBM (ours)** | $\mathcal{O}(T + \tau)$ | $\mathcal{O}(C(T + \tau))$ | 987K |

## E    CRPS RESULTS FOR LONG-TERM FORECASTS

In this section, we evaluate the long-term forecasting capabilities of our model using the CRPS metric and compare its performance against three established models: DeepAR (Salinas et al., 2020), Deep State Space (Rangapuram et al., 2018) and TimeGrad (Rasul et al., 2021), a similar EBM-based model. We excluded RSGLS-ISSM and ARSGLS models (Kurle et al., 2020) from Table 2 because their code implementations were not publicly available. Additionally, we omitted the Wikipedia dataset due to its limited length of 792 time steps, which is insufficient for the 336 and 720 forecast horizons. For all models, we employed a context length of 96. The results are summarized in Table 8. TF-EBM consistently outperforms the other models on the electricity and traffic datasets, demonstrating its superior long-term forecasting capabilities. While it ranks second on the exchange rate dataset, its overall performance highlights its effectiveness in long-horizon probabilistic forecasting tasks.

Table 8: Additional CRPS metric (lower is better) results for long-term foreacasting of 336 and 720 time steps. Mean and std. deviation are computed over 3 independent runs of each method. The best results are highlighted in red.

| Models | exchange rate | | electricity | | traffic | |
|---|---|---|---|---|---|---|
| | 336 | 720 | 336 | 720 | 336 | 720 |
| DeepAR | 0.030±0.001 | 0.065±0.008 | 0.092±0.001 | 0.120±0.000 | 0.243±0.005 | 0.265±0.005 |
| Deep State | 0.021±0.000 | 0.040±0.000 | 0.090±0.001 | 0.110±0.001 | 0.231±0.001 | 0.262±0.001 |
| TimeGrad | 0.053±0.010 | 0.079±0.009 | 0.078±0.007 | 0.087±0.017 | 0.203±0.003 | 0.313±0.023 |
| **TF-EBM (ours)** | 0.028 ±0.001 | 0.047±0.003 | 0.075±0.001 | 0.079±0.002 | 0.143±0.003 | 0.245±0.008 |

## F    TRANSFER LEARNING

This section evaluates our model's transfer learning capabilities. Specifically, we pre-train our model on the electricity dataset and then fine-tune it on the traffic and four ETT datasets using only three epochs, achieving a substantial reduction in computational time. As evident from Table 9, the fine-tuned model outperforms or matches the performance of other models. Additionally, the fine-tuning performance is generally lower than that of self-supervised training. This is expected, as the fine-tuning process utilizes a shorter training period and the model is not explicitly optimized for each dataset. However, the fine-tuning results on the traffic dataset surpass those of self-supervised training. This suggests that the time and frequency features extracted by TF-EBM from both the electricity and traffic datasets share similarities, allowing the model to effectively transfer knowledge between the two domains.

Table 9: Transfer learning task. MSE/MAE (lower is better) results for different prediction lengths and the same lookback window length 96. TF-EBM is pre-trained on the electricity dataset and the model is transfered to other datasets. Results of all other methods are taken from Wu et al. (2022). The best result is highlighted in red.

| models | | TF-EBM (ours) | | | | TimesNet | | Dlinear | | FEDformer | | Autoformer | |
|---|---|---|---|---|---|---|---|---|---|---|---|---|---|
| | | Fine-tuning | | Self-supervised | | | | | | | | | |
| metric | | MSE | MAE | MSE | MAE | MSE | MAE | MSE | MAE | MSE | MAE | MSE | MAE |
| ETTh1 | 96 | 0.395 | 0.408 | 0.382 | 0.400 | 0.384 | 0.402 | 0.386 | 0.400 | 0.376 | 0.419 | 0.449 | 0.459 |
| | 192 | 0.445 | 0.432 | 0.437 | 0.433 | 0.436 | 0.429 | 0.437 | 0.432 | 0.420 | 0.448 | 0.500 | 0.482 |
| | 336 | 0.526 | 0.491 | 0.472 | 0.450 | 0.491 | 0.469 | 0.481 | 0.459 | 0.459 | 0.465 | 0.521 | 0.496 |
| | 720 | 0.469 | 0.463 | 0.479 | 0.477 | 0.521 | 0.500 | 0.519 | 0.516 | 0.506 | 0.507 | 0.514 | 0.512 |
| ETTh2 | 96 | 0.297 | 0.358 | 0.297 | 0.349 | 0.340 | 0.374 | 0.333 | 0.387 | 0.358 | 0.397 | 0.346 | 0.388 |
| | 192 | 0.389 | 0.401 | 0.368 | 0.400 | 0.402 | 0.414 | 0.477 | 0.476 | 0.429 | 0.439 | 0.456 | 0.452 |
| | 336 | 0.428 | 0.434 | 0.411 | 0.433 | 0.452 | 0.452 | 0.594 | 0.541 | 0.496 | 0.487 | 0.482 | 0.486 |
| | 720 | 0.428 | 0.470 | 0.428 | 0.453 | 0.462 | 0.468 | 0.831 | 0.657 | 0.463 | 0.474 | 0.515 | 0.511 |
| ETTm1 | 96 | 0.355 | 0.381 | 0.339 | 0.374 | 0.338 | 0.375 | 0.345 | 0.372 | 0.379 | 0.419 | 0.505 | 0.475 |
| | 192 | 0.399 | 0.408 | 0.377 | 0.396 | 0.374 | 0.387 | 0.380 | 0.389 | 0.426 | 0.441 | 0.553 | 0.496 |
| | 336 | 0.439 | 0.433 | 0.432 | 0.432 | 0.410 | 0.411 | 0.413 | 0.413 | 0.445 | 0.459 | 0.621 | 0.537 |
| | 720 | 0.494 | 0.463 | 0.470 | 0.456 | 0.478 | 0.450 | 0.474 | 0.453 | 0.543 | 0.490 | 0.671 | 0.561 |
| ETTm2 | 96 | 0.183 | 0.272 | 0.176 | 0.266 | 0.187 | 0.267 | 0.193 | 0.292 | 0.203 | 0.287 | 0.255 | 0.339 |
| | 192 | 0.248 | 0.314 | 0.237 | 0.301 | 0.249 | 0.309 | 0.284 | 0.362 | 0.269 | 0.328 | 0.281 | 0.340 |
| | 336 | 0.316 | 0.355 | 0.302 | 0.342 | 0.321 | 0.351 | 0.369 | 0.427 | 0.325 | 0.366 | 0.339 | 0.372 |
| | 720 | 0.428 | 0.425 | 0.398 | 0.403 | 0.408 | 0.403 | 0.554 | 0.522 | 0.421 | 0.415 | 0.433 | 0.432 |
| traffic | 96 | 0.447 | 0.295 | 0.542 | 0.342 | 0.593 | 0.321 | 0.650 | 0.396 | 0.587 | 0.366 | 0.613 | 0.388 |
| | 192 | 0.459 | 0.301 | 0.539 | 0.353 | 0.617 | 0.336 | 0.598 | 0.370 | 0.604 | 0.373 | 0.616 | 0.382 |
| | 336 | 0.475 | 0.305 | 0.529 | 0.335 | 0.629 | 0.336 | 0.605 | 0.373 | 0.621 | 0.383 | 0.622 | 0.337 |
| | 720 | 0.496 | 0.320 | 0.534 | 0.346 | 0.640 | 0.350 | 0.645 | 0.394 | 0.626 | 0.382 | 0.660 | 0.408 |

## G  UNCERTAINTY ESTIMATION

Our TF-EBM model's uncertainty quantification capability stems from its probabilistic nature. As a probabilistic time series model, TF-EBM generates probability distributions for future values, allowing us to sample from these distributions to obtain the empirical quantiles of the uncertainty of our predictions. This probabilistic approach enables us to readily quantify the uncertainty associated with our predictions.

Leveraging the probabilistic nature of the TF-EBM model, we conducted uncertainty estimation experiments using synthetic data. As illustrated by the red line in Figure 5, the synthetic data was infused with varying levels of noise. As the variance of the injected noise increased, the observed curve exhibited more pronounced fluctuations. To demonstrate the model's ability to capture these variations, we injected high-variance noise into the curve (red line) prior to June 4, 2021, and minimal noise thereafter. As evident from the figure, the estimated 99% confidence interval in the region with minimal noise injection is narrower compared to the region with high-noise injection. This observation aligns with the uncertainty estimation provided by the TF-EBM model, demonstrating its ability to effectively capture uncertainty in the data.

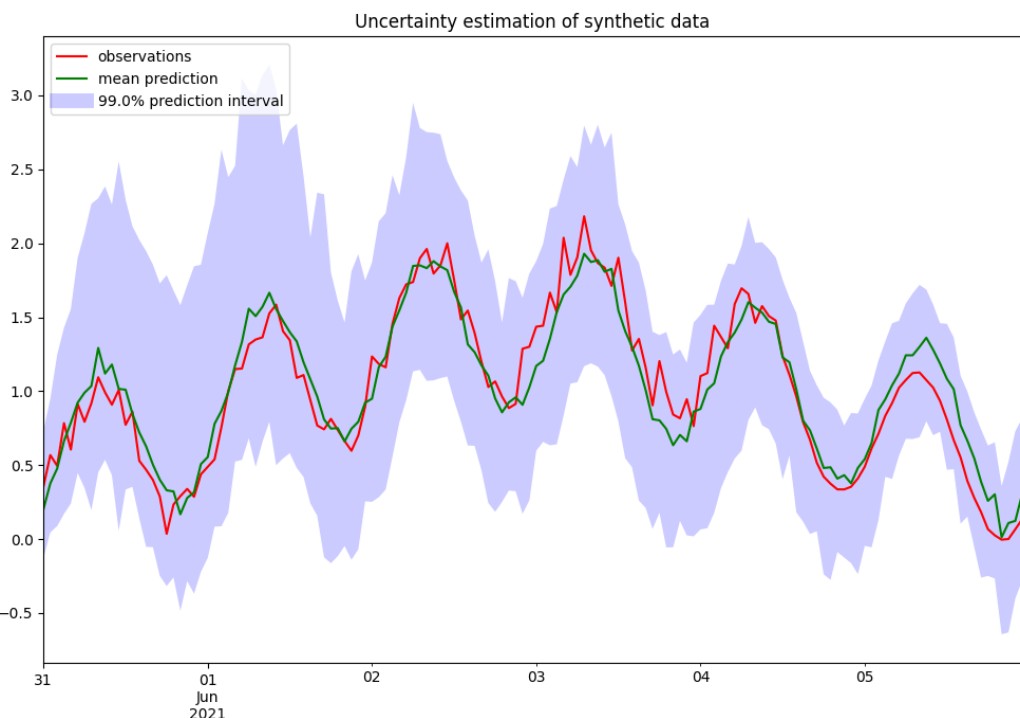

Figure 5: Uncertainty estimation of the TF-EBM model. Plot of prediction intervals for synthetic data using the TF-EBM model. The ground truth is represented by the red line, that is injected with varying levels of noise. The mean prediction is represented by the green line. High-variance noise is injected into the red curve before June 4, 2021, while minimal noise is injected after that date.

