# OpenReview forum: "GENERATIVE TIME SERIES LEARNING WITH TIME-FREQUENCY FUSED ENERGY-BASED MODEL"
_ICLR.cc/2024/Conference — Submitted to ICLR 2024_

### Official Review · Reviewer_DVHZ · 2023-10-26

**Soundness:** 3 good
**Presentation:** 2 fair
**Contribution:** 3 good
**Rating:** 5
**Confidence:** 4

**Summary:**

They proposed an energy-based model capable of both time-series forecasting and imputation. The model consists of a residual dilated convolutional network and considers both time and frequency features. They show good performance in long-term time series forecasting and imputation.

**Strengths:**

Originality: Models that apply energy-based models to time-series tasks already exist, but the proposed model (TB-EBM) differs in that it maintains long-term coherence better.

Quality: They structured the introduction, related works, and proposed method well to show the differences between TB-EBM and existing models.

Clarity: The paper is expressed well.

Significance: It is very interesting that it can handle long-term time-series forecasting and imputation at the same time.

**Weaknesses:**

1. They explained their model well, but the placement of Figures and Tables is not appropriate.
2. Figure 1 is not intuitive. I don't know what the "+" mark next to the Time-Freq Block part means. Also, the inference part in Figure 1 is more difficult to understand.
3. The table mentioned in the ablation study is difficult to see because it is not in the main paper.
4. The overall structure of the paper is not friendly to readers.

**Questions:**

1. Isn't there a process in Equation 6 to invert from the fequency domain back to the time domain? If so, I don't understand how the fequency domain feature and time domain feature are added in Equation 7.

2. The residual dilated convolutional network part in the 4.4 ablation study does not seem to have any meaning. There is no doubt that performance increases as the number of layers increases because the capacity of the model increases. Rather, it seems meaningful to use the same number of layers but subtract the residual part.

3. In [1], only one linear layer shows better performance than all existing models in long-term time-series forecasting. A model comparison with [1] seems necessary.

[1] Li, Zhe, et al. "Revisiting Long-term Time Series Forecasting: An Investigation on Linear Mapping." arXiv preprint arXiv:2305.10721 (2023).

---

> ### Author Response · Authors · 2023-11-18
> **Response to Reviewer DVHZ(Part1)**
>
> We thank the reviewer for offering the valuable feedback. We have addressed each of the concerns raised by the reviewer as below.
>
> W1.They explained their model well, but the placement of Figures and Tables is not appropriate.
>
> ● We rearranged some tables and figures in the updated paper .
>
> W2.Figure 1 is not intuitive. I don't know what the "+" mark next to the Time-Freq Block part means. Also, the inference part in Figure 1 is more difficult to understand.
>
> ● We updated both panels of Figure 1 to make them clearer in the revised paper.
>
> ● The "+" mark means we add the time and freq feature as indicated in Equation (7).
>
> ● The Inference part at the bottom panel in Figure 1 is just a standard Langevin MCMC process, which shows how to sample one time step from $y_{1:T}$to $y_{T+1:T+\tau}$with 59 steps of Langevin sampling.
>
> W3.The table mentioned in the ablation study is difficult to see because it is not in the main paper.
>
> ● This is unfortunate due to the space limit of the main paper.
>
> W4.The overall structure of the paper is not friendly to readers.
>
> ● We rearranged some tables and figures to make the updated paper more reader friendly. However, we were unable to include all tables and figures to the main paper due to the space limit imposed by ICLR.
>
> Q1.Isn't there a process in Equation 6 to invert from the fequency domain back to the time domain? If so, I don't understand how the fequency domain feature and time domain feature are added in Equation 7.
>
> ● No, we do not invert the frequency domain back to the time domain. As the goal of the training the energy based model is to get well estimation of the unnormalized probatlity density for temoral path,  we extract the time domain and freqency domain feature for each layer, then add the time domain and frequency domain features, which yields better results in our experiments that concating them together.
>
> Q2.The residual dilated convolutional network part in the 4.4 ablation study does not seem to have any meaning. There is no doubt that performance increases as the number of layers increases because the capacity of the model increases. Rather, it seems meaningful to use the same number of layers but subtract the residual part.
>
> ● We try to show in this ablation study that the multi-layer rather than a single-layer dilated convolutional neural network is necessary to filter out the noise in the time series. It is well-known that some high frequency modes could be noise instead of valuable signal. Since our TF-EBM model keeps all the frequency modes out of the FFT, which is different from some other works (FedFormer, Fast Fourier Convolution, etc. ) that keeps a portion of them, we reply on such multi-layer dilated CNN to automatically extract useful modes.

---

> > ### Author Response · Authors · 2023-11-18
> > **Response to Reviewer DVHZ(Part2)**
> >
> > Q3.In [1], only one linear layer shows better performance than all existing models in long-term time-series forecasting. A model comparison with [1] seems necessary.
> >
> > ● We added RLinear model results on the seven public datasets as in following table. We used the official pytorch implemention by the authors: https://github.com/plumprc/RTSF. We set context length to 96 to be the same as our model setting. Other hyperparameters remain the same as in the "script.md" file.
> >
> > ● As one can see, the RLinear model does performs better in terms of both MSE and MAE metric for ETTh1 and ETTh2 datasets.  However, our TF-EBM still outperforms when the prediction length is 720 for ETTh2. For ETTm1 and ETTm2, RLinear performs better in terms of MAE for most of the prediction lengths, while our TF-EBM model performs better in terms of MSE. However, TF-EBM performs better in terms of both MSE and MAE for the exchange, electriciy, and traffic datasets. Overall, the results demonstrate TF-EBM's long-term forecasting capability compared with the RLinear model.
> >
> > [1] Li Z, Qi S, Li Y, et al. Revisiting Long-term Time Series Forecasting: An Investigation on Linear Mapping[J]. arXiv preprint arXiv:2305.10721, 2023.
> >
> > |               |             | RLinear |        | TF-EBM |       |
> > |---------------|-------------|---------|--------|--------|-------|
> > | Dataset       | Pred Length | MSE     | MAE    | MSE    | MAE   |
> > | ETTh1         | 96          | 0.381   | 0.393  | 0.382  | 0.4   |
> > |               | 192         | 0.434   | 0.421  | 0.437  | 0.433 |
> > |               | 336         | 0.4709  | 0.4377 | 0.472  | 0.45  |
> > |               | 720         | 0.4673  | 0.4606 | 0.479  | 0.477 |
> > | ETTh2         | 96          | 0.2788  | 0.3336 | 0.297  | 0.349 |
> > |               | 192         | 0.3608  | 0.3878 | 0.368  | 0.4   |
> > |               | 336         | 0.3796  | 0.4104 | 0.411  | 0.433 |
> > |               | 720         | 0.4317  | 0.4456 | 0.428  | 0.453 |
> > | ETTm1         | 96          | 0.3515  | 0.3692 | 0.339  | 0.374 |
> > |               | 192         | 0.3886  | 0.3866 | 0.377  | 0.396 |
> > |               | 336         | 0.4203  | 0.4075 | 0.432  | 0.432 |
> > |               | 720         | 0.4787  | 0.4402 | 0.47   | 0.456 |
> > | ETTm2         | 96          | 0.1824  | 0.2653 | 0.176  | 0.266 |
> > |               | 192         | 0.2477  | 0.3059 | 0.237  | 0.301 |
> > |               | 336         | 0.3092  | 0.3431 | 0.302  | 0.342 |
> > |               | 720         | 0.4055  | 0.3977 | 0.398  | 0.403 |
> > | Exchange Rate | 96          | 0.0827  | 0.2007 | 0.085  | 0.205 |
> > |               | 192         | 0.1799  | 0.3002 | 0.171  | 0.296 |
> > |               | 336         | 0.3463  | 0.4226 | 0.287  | 0.391 |
> > |               | 720         | 0.9143  | 0.7181 | 0.843  | 0.699 |
> > | Electricity   | 96          | 0.1972  | 0.2736 | 0.173  | 0.265 |
> > |               | 192         | 0.1965  | 0.2762 | 0.186  | 0.278 |
> > |               | 336         | 0.2113  | 0.2916 | 0.198  | 0.289 |
> > |               | 720         | 0.2533  | 0.3248 | 0.235  | 0.319 |
> > | Traffic       | 96          | 0.6579  | 0.3916 | 0.542  | 0.342 |
> > |               | 192         | 0.5994  | 0.3632 | 0.539  | 0.353 |
> > |               | 336         | 0.602   | 0.3649 | 0.529  | 0.335 |
> > |               | 720         | 0.6424  | 0.3861 | 0.534  | 0.346 |

---

> > > ### Author Response · Authors · 2023-11-22
> > > **Feedback Request**
> > >
> > > Dear reviewer DVHZ:
> > >
> > > With the author/reviewer discussions concluding in just a day, we would be grateful if you could confirm whether our response adequately addresses your primary concerns. If so, we respectfully request that you reconsider the score.
> > >
> > > Your further guidance on the paper and/or our rebuttal would be highly valued. We remain open to further discussion and paper refinement. To ensure ample time to address any lingering or new questions, we kindly request that our next round of communication be scheduled accordingly.
> > >
> > > Thank you sincerely for your dedication to enhancing our work on time series learning.

---

> > > > ### Comment · Reviewer_DVHZ · 2023-11-22
> > > >
> > > > Thank you for the response to my questions.
> > > >
> > > > First of all, I am grateful that the paper was revised to be much easier to understand than the previous paper. But I still have two questions.
> > > >
> > > > 1) In Equation 7, the scales of the frequency domain value and the time domain value will be different. Won’t one dominate?
> > > >
> > > > 2) Why does TF-EBM perform better in the long-range?

---

> > > > > ### Author Response · Authors · 2023-11-22
> > > > > **Response to Reviewer DVHZ**
> > > > >
> > > > > Q1:In Equation 7, the scales of the frequency domain value and the time domain value will be different. Won’t one dominate?
> > > > >
> > > > > ● We performed normalization on our input time series to ensure consistant scale along the temporal dimension. Prior to fusing the extracted time and frequency features, we verified that they did not exhibit significant differences in terms of scale. Additionally, the ablation study C.2 revealed that the model equipped with both feature extraction modules outperformed those equipped with only time or frequency features across various datasets and forecasting ranges. This finding underscores the importance of both time and frequency features for effective time series modeling.
> > > > >
> > > > > Q2.Why does TF-EBM perform better in the long-range?
> > > > >
> > > > > ● TF-EBM excels in long-term forecasting due to its distinct modeling approach. Unlike autoregressive models that generate forecasts one step at a time, TF-EBM generates multi-step forecasts simultaneously, effectively circumventing the error accumulation issue that plagues autoregressive models. Additionally, TF-EBM explicitly models the temporal dependency by capturing the joint distribution of the entire temporal sequence, a capability that MLP-based models lack.
> > > > >
> > > > > ● Furthermore, TF-EBM extracts both time and frequency features from the time series, providing a more comprehensive representation of the data. Frequency features are particularly adept at capturing long-term patterns, and their inclusion in TF-EBM's modeling framework significantly enhances its long-term forecasting performance.

---

### Official Review · Reviewer_op54 · 2023-10-29

**Soundness:** 3 good
**Presentation:** 3 good
**Contribution:** 3 good
**Rating:** 6
**Confidence:** 2

**Summary:**

The paper proposes a generative model for time series forecasting (and imputation). The proposed energy based model makes use of both temporal and frequency based features through a neural network architecture the paper calls Time-Frequency block. The two sub-parts that make up this block are derived from previous works but the combination allows for integrating local and global patterns.

**Strengths:**

The paper makes clever use of existing work in time series feature extraction to propose the Time-Frequency block as an original neural network building block for time series data. It would in fact be curious to see how an NN based on these blocks works for other more general tasks around time-series data such as prediction, unsupervised learning etc.

The paper is overall well written and is clear in its descriptions and in providing relevant background. The intro and related works section does a good job at summarizing the paper as well as how the proposed method relates to and improves upon existing approaches.

The proposed method is directed towards the significant task of time series forecasting, and the results seem promising.

**Weaknesses:**

This is overall a good paper. I do however have some concerns around the experiments section which are detailed in Questions below:

**Questions:**

1: It's unclear how significant the resulting improvements are, the numbers. Can the authors quantify if the improvements are significant? (instead of just reporting means, report the errors too). Also, this might be a typo but Table 3 first row FEDformer MSE is the lowest.

2: It could be worth discussing why certain differences in results arise, e.g. Table 2 DeepAR seems to perform basically identical to the proposed method for exhange rate and electricity datasets, but performs noticeably worse on the other two datasets. What's the reason? This could provide insight into where the proposed method can provide maximum improvement (and why).

3: The paper mentions that deterministic methods generally don't help with uncertainty quantification, that sets the reader up to seeing uncertainty quantification being addressed by the paper, but it doesn't seem like it has been addressed in text or in experiments.

I'll be happy to revisit my score if the above points are meaningfully addressed.

---

> ### Author Response · Authors · 2023-11-18
> **Response to Reviewer op54**
>
> We thank the reviewer for offering the valuable feedback. We have addressed each of the concerns raised by the reviewer as below.
>
> Q1: It's unclear how significant the resulting improvements are, the numbers. Can the authors quantify if the improvements are significant? (instead of just reporting means, report the errors too). Also, this might be a typo but Table 3 first row FEDformer MSE is the lowest.
>
> ● Our TF-EBM model, being a probabilistic time series model, is evaluated using the CRPS metric to assess its distribution modeling capabilities. To strengthen the comparison, we have incorporated the TimeGrad model, another prominent EBM-based time series model, into Table 1. Additionally, we have included extended CRPS results for longer forecast horizons of 336 and 720 time steps in Table 8 of the Appendix. The improvements over state-of-the-art probabilistic models can be substantial, with a 30% reduction in CRPS for traffic data. Furthermore, we present both MAE and MSE metrics based on the error between the mean value of the forecasted distribution and the ground truth. Employing the mean value as the point estimate allows for a fair comparison with recent transformer-based models.
>
> ● The typo in the MSE field inTable 3 for FEDformer is fixed in the updated paper.
>
> Q2: It could be worth discussing why certain differences in results arise, e.g. Table 2 DeepAR seems to perform basically identical to the proposed method for exhange rate and electricity datasets, but performs noticeably worse on the other two datasets. What's the reason? This could provide insight into where the proposed method can provide maximum improvement (and why).
>
> ● The exchange rate and electricity datasets contain 8 and 370 time series, respectively, while the traffic and Wikipedia datasets have 963 and 2000 series, significantly larger than the other two datasets. The exchange rate and electricity datasets have relatively high autocorrelations (>90% as shown in Table 6) which could make them suitable for autoregressive models such as DeepAR, especially in short-horizon forecasting.
>
> ● We performed additional experiments by extending the forecast horizon to 336 and 720 time steps, using the CRPS metric to evaluate performance. As shown in Table 8 in the Appendix of the updated paper, our TF-EBM outperforms DeepAR by a substantial margin for the electricity and exchange rate datasets, except for the electricity dataset with a 336 forecast horizon. This demonstrates TF-EBM's superior long-term forecasting capability, while DeepAR struggles with error accumulation
>
> Q3: The paper mentions that deterministic methods generally don't help with uncertainty quantification, that sets the reader up to seeing uncertainty quantification being addressed by the paper, but it doesn't seem like it has been addressed in text or in experiments.
>
> ● Our TF-EBM model's uncertainty quantification capability stems from its probabilistic nature. As a probabilistic time series model, TF-EBM generates probability distributions for future values, allowing us to sample from these distributions to obtain the empirical quantiles of the uncertainty of our predictions. This probabilistic approach enables us to readily quantify the uncertainty associated with our predictions. For example, we can output the 70% confidence interval along with the mean value as shown in Figure 3 in the appendix. In contrast, point forecast models like TimesNet and AutoFormer only provide single-valued predictions for each time step, lacking the ability to capture the inherent uncertainty in time series data.
>
> ● To further demonstrate the uncertainty quantification capability, we performed an experiment in Section G in the appendix of the updated paper. The synthetic ground truth is injected with different levels of noise. We output the 99% confidence interval along with the mean forecasts. One can observe that the model forecasts in the time steps with larger noise variance show wider confidence interval than those in the low variance time stpe. The uncertainty of the forecasts is consisitent with the level of the noise.

---

> > ### Author Response · Authors · 2023-11-22
> > **Feedback Request**
> >
> > Dear reviewer op54:
> >
> > With the author/reviewer discussions concluding in just a day, we would be grateful if you could confirm whether our response adequately addresses your primary concerns. If so, we respectfully request that you reconsider the score.
> >
> > Your further guidance on the paper and/or our rebuttal would be highly valued. We remain open to further discussion and paper refinement. To ensure ample time to address any lingering or new questions, we kindly request that our next round of communication be scheduled accordingly.
> >
> > Thank you sincerely for your dedication to enhancing our work on time series learning.

---

> > > ### Comment · Reviewer_op54 · 2023-11-22
> > > **Response**
> > >
> > > Thank you for providing this response. Re the first point, what I was asking for is, e.g. table 3 the results are the mean over 5 independent runs. That doesn't give me any information about how those those runs did, incorporating the standard deviation over those 5 runs can provide better insight.

---

> > > > ### Author Response · Authors · 2023-11-23
> > > > **Response to Reviewer op54**
> > > >
> > > > Q1:Thank you for providing this response. Re the first point, what I was asking for is, e.g. table 3 the results are the mean over 5 independent runs. That doesn't give me any information about how those those runs did, incorporating the standard deviation over those 5 runs can provide better insight.
> > > >
> > > > ● Thank you for your valuable feedback. We acknowledge the importance of evaluating model performance using the standard deviation.
> > > >
> > > > While we primarily followed the conventions established in previous papers, we recognize the need to incorporate this measure into our analysis. The CRPS metric was accompanied by the standard deviation in previous probabilistic models, while point estimate models reporting MAE/MSE results typically omitted it. Unfortunately, due to time constraints as the author/reviewer discussions are nearing completion, we are unable to re-run all other models to obtain their standard deviations for comparison. Consequently, even if we were to include the standard deviation for our model, it would lack a baseline for comparison.

---

### Official Review · Reviewer_yqBZ · 2023-11-03

**Soundness:** 3 good
**Presentation:** 3 good
**Contribution:** 3 good
**Rating:** 5
**Confidence:** 4

**Summary:**

The paper proposes a novel generative model called Time-Frequency fused Energy-Based Model (TF-EBM) for long-term probabilistic time series forecasting and imputation.
TF-EBM is an encoder-only model that employs energy-based learning to construct an unnormalized probability density over temporal paths. This allows coherent long-term forecasting.

**Strengths:**

Originality:

Proposes a novel architecture combining energy-based models and time-frequency modeling for time series, which is an original contribution.

Leverages energy-based learning in a new way for coherent long-term time series forecasting.

Pre-training approach for time series using TF-EBM is an original idea inspired by NLP models.

Quality:

Comprehensive experiments across various forecasting and imputation tasks on multiple datasets.

Comparison to many strong baselines like DeepAR, Autoformer, etc. demonstrates quality.

Strong performance especially on long-term forecasting shows effectiveness of the approach.

Ablation studies analyze the model components like dilated CNN and time-frequency modules.

Clarity:

The method and architecture are clearly explained with useful diagrams.

The related work section covers relevant literature on energy-based models, transformers, etc.

Experiments are well-organized and different tasks nicely showcase model capabilities.

**Weaknesses:**

The motivation for using energy-based learning specifically is not clearly articulated. References connecting energy-based models to time series properties could help.

More analysis on why the time-frequency modeling outperforms just time or just frequency features could strengthen this contribution.

The pre-training evaluation is limited. More exhaustive experiments on the transfer learning capabilities could be done.

Only univariate time series are evaluated. Comparing with PatchTST is needed in this setting.

The comparison to autoregressive models like LSTMs is missing. This could reveal advantages over common recurrent approaches.

The synthetic ablation study focuses on noise removal. Ablations on modeling long-term dependencies could be more insightful.

All datasets are regular time series. Applying to irregularly sampled data from healthcare etc. could reveal robustness.

Uncertainty estimation and calibration are not evaluated for the probabilistic forecasting. This could be an issue.

Hyperparameter tuning details are not provided clearly. It's unclear if suboptimal settings affect comparisons.

The advantages over previous energy-based time series methods like TimeGrad are not fully fleshed out.

**Questions:**

See the weakness.

**Details Of Ethics Concerns:**

-

---

> ### Author Response · Authors · 2023-11-18
> **Response to Reviewer yqBZ(Part1)**
>
> W1.The motivation for using energy-based learning specifically is not clearly articulated. References connecting energy-based models to time series properties could help.
>
> ● Energy based model (EBM) has been widely applied in the image generation because EBM models high-dimentional joint distribution via a flexible unnormalized probability density benefitting deep model design. As for the time series, forecasting task, especailly for long-term forecast needs to model the long-term temporal coherence, and EBM can naturally serve as a powerful tool to model the joint distribution of the long-term correlation. There are limited number of papers applying EBM on time series forecasting works to the best of our knowledge. Previous EBM models used in time series forecasting including Time Grad/Score grad cited in the Related work section focus on modeling the join distribution across time series at each time step, while we utilize the EBM to model the joint distribution along the temporal path of the time series, which can guarantee the maximum likelihood of the whole sequence.
>
> W2.More analysis on why the time-frequency modeling outperforms just time or just frequency features could strengthen this contribution.
>
> ● We consider the particularity of time series as seqence data which can be descirbed in the time and freqency demension. As we laid out in Section 3.5 and references therein, the time and frequency dimension mostly characterize the local and global patterns of the time series, respectivley, that are both crucial for long-term time series forecasting. Therefore, it is a natural choice to combine both patters to enhance the model performance.
>
> ● We added additinoal ablation study on the exchange rate and traffic datasets with results included in Table 6 in the Appendix. The results on the full datasets show that combing both time and frequency features contribute better forecasting ability in our model design.
>
> W3.The pre-training evaluation is limited. More exhaustive experiments on the transfer learning capabilities could be done.
>
> ● We evaluated the transfer learning capabilities of our model by pre-training it on the electricity dataset and fine-tuning it on the traffic and four ETT datasets. The results, summarized in Table 9, demonstrate that the fine-tuned model outperforms or matches the performance of other models.
>
> W4.Only univariate time series are evaluated. Comparing with PatchTST is needed in this setting.
>
> ● As we mentioned in both the Introduction and the model framework section, unlike PatchTST, TF-EBM is an univaraite time-series model, and we will extend it to the multivaraite case as we mentioned in the Future work section. We wil compare with PathTST by that time.
>
> W5.The comparison to autoregressive models like LSTMs is missing. This could reveal advantages over common recurrent approaches.
>
> ● We showed DeepAR model results in Table 1 for the CRPS metric. DeepAR is a strong autoregressive model based on LSTM. Additionally, in both Table 1 and 8, we added results from the TimeGrad model, which is a strong autoregressive EBM model.
>
> ● We appended additional column in Table 3 for the MAE/MSE metric obtained using LSTM model. They are also taken directly from TimesNet paper.
>
> ● TF-EBM outperforms both DeepAR and TimeGrad on most of the datasets.

---

> > ### Author Response · Authors · 2023-11-18
> > **Response to Reviewer yqBZ(Part2)**
> >
> > W6.The synthetic ablation study focuses on noise removal. Ablations on modeling long-term dependencies could be more insightful.
> >
> > ● We added additional ablation studies for long-term forecasts in Figure 4 in the Appendix.  Long-term pattern mainly construct low frequency components in time series. In the ablation study, we forecast the future time steps using one layer, two layers and three layers of the Time-Freq Block separately and apply the FFT for the different forecast. The experiment results show that three layers of Time-Freq Block can well charaterize the low-frequency part and indicate that stack of Time-Freq Block help to capture long term pattern.
> >
> > W7.All datasets are regular time series. Applying to irregularly sampled data from healthcare etc. could reveal robustness.
> >
> > ● TF-EBM is a time series model that is currently designed for synchronous time series data. However, there are several approaches to dealing with irregularly sampled or asynchronous time series data, such as converting the asynchronous timestamps to synchronous ones by binning. This approach could be implemented in future work to expand the capabilities of TF-EBM.
> >
> > W8.Uncertainty estimation and calibration are not evaluated for the probabilistic forecasting. This could be an issue.
> >
> > ● We add the text description and uncertainty experiments in Section G in the appendix of the updated paper. We inject different level of noise into the synthetic data, and observe that the forecast  of our model in the region with bigger variance of the noise injection show wider 99% internal than the forecast in region with little variance of the noise injection. The estimation of the uncertainty is in consisitent with the level of the noise. This probabilistic approach enables us to readily quantify the uncertainty associated with our predictions.
> >
> > W9.Hyperparameter tuning details are not provided clearly. It's unclear if suboptimal settings affect comparisons.
> >
> > ● Other model results in the experiments were directly taken from the paper TimesNet, i.e., we did not run those models. We just used the same training and testing sets for each dataset for a fair comparison. We will release our code implenentation with hyperpameter settings for TF-EBM once the paper is accepted.
> >
> > W10.The advantages over previous energy-based time series methods like TimeGrad are not fully fleshed out.
> >
> > ● In contrast to TimeGrad, which is an autoregressive EBM model and therefore inefficient for long-term forecasts, our TF-EBM model is well-suited for long-horizon predictions. To quantitatively compare both models, we replicated the experiments from Table 1 using TimeGrad and evaluated the models based on CRPS measures. Additionally, due to space constraints, we included the results of long-term probabilistic forecasting in Table 5 of the Appendix. Note that we modified the format of Table 1 to enhance its compactness for the main paper.
> >
> > ● Experimental results indicate that TF-EBM outperforms TimeGrad on nearly all datasets, with the exception of the rolling 30-day forecasts for the exchange rate dataset. This exception is likely due to the relatively small size of the exchange rate dataset, consisting of only 8 time series, and its characteristics of high autocorrelation and low standard deviation. These characteristics may make the exchange rate dataset more suitable for autoregressive models when the forecast horizon is short.

---

> > > ### Author Response · Authors · 2023-11-22
> > > **Feedback Request**
> > >
> > > Dear reviewer yqBZ:
> > >
> > > With the author/reviewer discussions concluding in just a day, we would be grateful if you could confirm whether our response adequately addresses your primary concerns. If so, we respectfully request that you reconsider the score.
> > >
> > > Your further guidance on the paper and/or our rebuttal would be highly valued. We remain open to further discussion and paper refinement. To ensure ample time to address any lingering or new questions, we kindly request that our next round of communication be scheduled accordingly.
> > >
> > > Thank you sincerely for your dedication to enhancing our work on time series learning.

---

> > > > ### Comment · Reviewer_yqBZ · 2023-11-23
> > > > **Thanks for the response**
> > > >
> > > > Dear Authors,
> > > >
> > > > Thanks for your efforts. Most of my concerns are addressed. My remaining question is that: in relation to the W4, it should be noted that the majority of recent methods primarily accept univariate time-series data as input. These methods typically adhere to the principle of channel independence when modeling multivariate time series. Therefore I would like stick my score at this stage.
> > > >
> > > > Best

---

> > > > > ### Author Response · Authors · 2023-11-23
> > > > > **Response to Reviewer yqBZ**
> > > > >
> > > > > Thank you for your feedback. There is another reason that we did not incorporate PatchTST in our comparison. The MAE/MSE metrics for all other models were directly extracted from the TimesNet paper, maintaining consistency in data sources, while the PatchTST results presented in the corresponding paper diverged from those obtained using the TimesNet paper when applied to the same dataset. Since we adhered to the data preparation process outlined in the TimesNet paper, we opted to exclude PatchTST from our comparison. We are unable to include them in this version due to time constraints, but we will endeavor to include PatchTST in our revised paper after replicating it using the same settings as TF-EBM.

---

### Official Review · Reviewer_5STj · 2023-11-03

**Soundness:** 2 fair
**Presentation:** 3 good
**Contribution:** 3 good
**Rating:** 6
**Confidence:** 2

**Summary:**

This manuscript describes a generative model that can be used for imputing or forecasting a univariate time series. The model's encoder block consists of a sequence of "time-freq blocks" (TFB). Within a single TFB, convolutional and MLP layers act on both the original input in the time domain, as well as the concatenated real and imaginary parts of a Fourier transform of those inputs. TFB are connected with residual connections. The decoder block consists of a series of convolutional layers followed by MLP layers. Model estimation was done via maximum likelihood by minimizing a measure of divergence between observed data samples and samples generated from the model -- though I will say that I am not familiar with the specific estimation methods used, and did not read them carefully. Experiments with forecasting and imputation tasks indicate that the model has performance that is generally comparable to or better than several recently published methods. An ablation study demonstrates that both the time and frequency features are useful.

**Strengths:**

My understanding is that the primary contribution of this article is the proposal of time-frequency blocks as described in the summary, which allow the model to use information from both the time and frequency domain. Previous modeling approaches have also incorporated mechanisms for incorporating time and frequency domain features, but the specific architecture proposed here may be novel. I do not possess sufficient knowledge of the related literature to fully assess the significance of this contribution, but it has the feeling of an interesting idea that is worth being published. The article was fairly clear overall; I could implement a similar model based on the descriptions given, though I could not exactly reproduce it. I am confident that man of my questions along these lines could be addressed in a revision.

**Weaknesses:**

The main weakness I see in the paper is that clear procedures for separating model development from the experiments evaluating model skill were not described. This leaves the reader with the impression that a clear "model development" and "model evaluation" data split was not made, and that claims about matching or exceeding state-of-the-art performance may not generalize to novel data sets. If careful procedures for evaluating the model on data that were not used for model development were indeed in place, it would be beneficial to describe them. If not, perhaps results could be given for new data sets?

**Questions:**

1. I found equations 6 and 7 to be helpful, but they did not seem to align precisely with Figure 1 and did not completely fill out all details of the model. A few specific questions are below:

    a) The text just above Eq 6 defines the operator $g = MLP(SiLU(Conv1D(\cdot)))$, while Figure 1 and earlier text indicate that Dilated Conv and Conv1d layers were used. Am I understanding correctly that the Dilated Conv and Conv layers in the figure represent the $g$ operator? If so, can this inconsistency be resolved? Otherwise, can the figure or text be amended to address this point of confusion?

    b) I find the layout of the sub-blocks within the TFB on the left side of Figure 1 confusing. The inputs to the DC -> SiLU -> Conv1D block are the time feats or frequency feats, right? This is not apparent to me from the organization of the figure, which has arrows pointing from the DC -> SiLU -> Conv1D block to the time/freq feat modules. And why is there an arrow pointing up from the Conv1D block?

    c) I believe from the figure that $y^l = y^{(l-1)} + TFB(y^{(l-1)} = y^{(l-1)} + f^l_{freq} + f^l_{time}$, but I am not fully confident in this and didn't see a clear statement of this in the text. It would be helpful to add a line to Eq 6 clearly stating how the output of a TFB is calculated.

2. How were missing data values (e.g., as in imputation settings) handled in computation of the FFT?

3. I don't understand the motivation for using a probabilistic score CRPS for a distributional forecast in the short-term forecasting problem, and the MAE/MSE for a point prediction in the long-term forecasting and imputation problems. It seems to me that it would be valuable to examine measures of point forecast skill and distributional forecast skill in both forecasting settings. This is particularly salient since the second sentence of the abstract reads, "However, existing models primarily focus on deterministic point forecasts, neglecting generative long-term probabilistic forecasting and pre-training models across diverse time series analytic tasks, which are essential for uncertainty quantification and computational efficiency." The abstract states that long-term probabilistic forecasting is an important problem, but we do not see CRPS results for the long-term forecasting example. Could a more complete suite of results be added?

4. The ablation study looking at the contribution of time and frequency extraction only looks at a subset of the data sets. I think it would be interesting to provide results for all data sets used in the paper, and to provide some examination to indicate settings in which it is more or less helpful to use both feature sets.  For example, could columns be added to Table 6 giving something like the lag 1 autocorrelation of the series and the power of the first few dominant frequencies? With the hypothesis being that the relative value of using both feature groups may depend on characteristics of the timer series being modeled?

---

> ### Author Response · Authors · 2023-11-18
> **Response to Reviewer 5STj**
>
> We thank the reviewer for offering the valuable feedback. We have addressed each of the concerns raised by the reviewer as below.
>
> W1.The main weakness I see in the paper is that clear procedures for separating model development from the experiments evaluating model skill were not described.
>
> ● The datasets we used to evaluate our model performance are all public datasets that are widely available and heavily tested in other time series forecasting literature. We used the same dataset splitting ratio as in other published papers for training and testing data.
>
> ● For the CRPS resutls in Table 2 and 8, we followed the same splitting ratio as in [1].
>
> ● For all the MAE/MSE results in Table 3, 4, 5, 6, and 9, we used the same data preparation code released by the authors of the TimesNet paper: https://github.com/thuml/Time-Series-Library/blob/main/data_provider/data_loader.py
>
> ● In order to evaluate model's performan on new datasets, we included the transfer learning experiments in the Appendix of the revised paper. To be specific, we pre-train our model on the electricity dataset, and fine tune it on traffic and four ETT datasets using just three epochs. The results are summarized in Table 9. The fine-tuning performance is still better than or comparable to other models.
>
> [1]Richard Kurle, Syama Sundar Rangapuram, Emmanuel de Bézenac, Stephan Günnemann, and Jan Gasthaus. Deep rao-blackwellised particle filters for time series forecasting.Advances in Neural Information Processing Systems, 33:15371–15382, 2020.
>
> Q1.I found equations 6 and 7 to be helpful, but they did not seem to align precisely with Figure 1 and did not completely fill out all details of the model. A few specific questions are below:
>
> ● a)The Dildated Conv and Conv layers shown inside the Time-Freq Block (TFB) in Figure 1 do not represent the g operator. Note that we named it as DSC (DilatedConv -> SiLU -> Conv1d) network in the updated paper to make it clearer. The g operator (g_time and g_freq) is only applied in the Time Feat Moudule and Frequency Feat Module in Figure 1 which is formalized in the Eq 6.  The Time-Freq Block shown in Figure 1 comprises two compents: one is the DSC network, and the other is the time-frequency (TF) domain features extracting network. The Figure 1, Eq.6 and Section 3.5 are all updated in the revised paper.
>
> ● b)The inputs to the DSC network shown in the sub-blocks within TFB on the left side of Figure 1 are not the time feats or frequency feats. The inputs to the DSC network of the L-th layer are the sum of the inputs and outputs of DSC of the  (L-1)th layer (defined by Eq 5 in the updated paper). The inputs to the time and frequency feats are the outputs of the DSC network (red arrow in Figure 1). The outputs of the time and frequency feats are added together to produce the ultimate energy scalar defined by Eq 7.
>
> ● c) We have named DilatedConv -> SiLU -> Conv1d as DSC in updated paper, so Eq 5 has been updated as
> $y^l=y^{l-1}+DSC(y^{l-1})$
> to clarify the statement.
>
> Q2.How were missing data values (e.g., as in imputation settings) handled in computation of the FFT?
>
> ● For both imputation and forecasting tasks, missing values are replaced with random Gaussian noise prior to the FFT transform, followed by Langevin sampling to generate model-derived samples.
>
> Q3.The abstract states that long-term probabilistic forecasting is an important problem, but we do not see CRPS results for the long-term forecasting example. Could a more complete suite of results be added?
>
> ● We included CRPS results for the long-term (forecast horizon equals 336 and 720) forecasting in Table 8 in the Appendix. We compared against three models: DeepAR, DeepState, and TImeGrad, but not the RSGLS-ISSM or ARSGLS because their code implementations were not publicly available. We did not test the Wiki dataset as it only has 792 data points that are not enough for the long-term forecast.
>
> Q4.The ablation study looking at the contribution of time and frequency extraction only looks at a subset of the data sets. I think it would be interesting to provide results for all data sets used in the paper.
>
> ● We performed the same ablation study on the exchange rate and traffic datasets as well. The results are included in Table 6 in the Appendix, which also includes the average of the lag-1 autocorrelation, standard deviation, and four dominant power spectrums. However, such characteristics of the timer series does not seem to have strong correlation with the relative value of using both feature groups.  Actually,the model equipped with time and frequency feature extracting module performs the best on almost all datasets and forecasting ranges compared with only equipped with either time or frequency extracting module. The three exceptions on the ETTm1 and traffic datasets do not seem to have strong statistical significance.

---

> > ### Author Response · Authors · 2023-11-22
> > **Feedback Request**
> >
> > Dear reviewer 5STj:
> >
> > With the author/reviewer discussions concluding in just a day, we would be grateful if you could confirm whether our response adequately addresses your primary concerns. If so, we respectfully request that you reconsider the score.
> >
> > Your further guidance on the paper and/or our rebuttal would be highly valued. We remain open to further discussion and paper refinement. To ensure ample time to address any lingering or new questions, we kindly request that our next round of communication be scheduled accordingly.
> >
> > Thank you sincerely for your dedication to enhancing our work on time series learning.

---

### Official Review · Reviewer_F51B · 2023-11-05

**Soundness:** 2 fair
**Presentation:** 2 fair
**Contribution:** 2 fair
**Rating:** 6
**Confidence:** 4

**Summary:**

This paper proposed a time-frequency fused parameterization of EBM for long-term time series modeling. The EBM defines a bottom-up mapping which is a encoder architecture as mentioned in this paper. The design of this encoder can be tricky. The author(s) leveraged a novel residual time-frequency block fuse information from time and frequency domain and hence got pretty good results in several benchmarks.

**Strengths:**

1. The proposed method is simple and clear.

2. The overall performance is good.

3. The masked pretraining method seems interesting and may be used to scale-up time series training.

**Weaknesses:**

The paper got good results in several benchmarks but lack of novelty in two ways. If the author claims to improve EBM based learning method in time series, the author should discuss more results about MCMC sampling results, e.g. the chain mixing problem. MCMC chain mixing is a well-known issue in non-sequential signals, it could be worse in time series. If the author claim the novel parameterization of EBM and specially designed fusion block (i.e. the important inductive bias for time series), I would suggest add more insightful ablation studies. There are many design choices for fusion. The author should clarify the benefits of current design choice.

Meanwhile, the author missed some pioneering literature about basic EBM, such as,

[1] "A theory of generative convnet." International Conference on Machine Learning. PMLR, 2016.

[2] "Implicit generation and modeling with energy based models." Advances in Neural Information Processing Systems 32 (2019).

[3] "Cooperative training of descriptor and generator networks." IEEE transactions on pattern analysis and machine intelligence 42.1 (2018): 27-45.

For short-run Langevin dynamics learning of EBM, [4] is also an important work.

[4] "A tale of two flows: Cooperative learning of langevin flow and normalizing flow toward energy-based model." ICLR (2022).

**Questions:**

See weakness.

---

> ### Author Response · Authors · 2023-11-18
> **Response to Reviewer F51B**
>
> We thank the reviewer for offering the valuable feedback. We have addressed each of the concerns raised by the reviewer as below.
>
> W1.The paper got good results in several benchmarks but lack of novelty in two ways. If the author claims to improve EBM based learning method in time series, the author should discuss more results about MCMC sampling results, e.g. the chain mixing problem. MCMC chain mixing is a well-known issue in non-sequential signals, it could be worse in time series. If the author claim the novel parameterization of EBM and specially designed fusion block (i.e. the important inductive bias for time series), I would suggest add more insightful ablation studies. There are many design choices for fusion. The author should clarify the benefits of current design choice.
>
> ● As explained in Section 3.2, we opted for the short-run MCMC technique due to its efficiency in accelerating the Langevin sampling process. Through our experimentation, we observed that even non-mixing chains have the capability to produce favorable outcomes, as demonstrated by the experiment results.
>
> ● Our main novelty is choosing EBM as the backbone to model the long-term temporal dependence of time series. The existing energy-based models applied in time series fields like Time Grad/Score Grad model the joint distribution of multiple time series at each time step, and use an autoregressive structure like RNN to model the temporal dependence which can suffer from error accumulation. However, we use EBM to model the joint distribution of the whole temporal path of each time series ( $y_{1:T+\tau}$) innovatively. During training and predicting process, we do not perform gradient update on the context$y_{1:T}$, but we only update the predicted values $y_{T+1:T+\tau}$following the Langevin dynamics with calculated gradient. Therefore, the forecasts $y_{T+1:T+\tau}$can be sampled at once benefiting the long-term forecasting, which is demonstrated in the extensive experiments.
>
> ● The main goal to applying EBM to time series modeling is to get a accurate estimator for the probability density of the entire whole temporal path, in which the time and frequency fusion architecture plays an important role. In the image generation work, the density estimation network is directly applied on the raw pixel, however, it is different in the time series field because time series can be described in both time and frequency dimension. Our novelty here is to use the DSC network defined in Section 3.4 of the updated paper to extract different levels of information of the time series. Specifically, we apply a time and frequency feat extracting network to extract local and global features, respectively, which are then added together to generate the ultimate energy scalar. Additionally, the ablation study conducted in Section 4.4 and Appendix C further demonstrates the effectiveness of our chosen modeling approach. It is worth noting that alternative fusion methods like concatenation or gating may potentially yield even better results, and therefore, in our future work, we plan to thoroughly investigate and explore these alternatives.
>
> ● In summary, our paper introduces a novel EBM-based encoder-only generative time series model that leverages fused time and frequency features.
>
>   ○ Our main novelty is to choose EBM as the backbone due to its flexibility as a general distribution modeling tool, as the energy function can be any scalar-returning function and does not need to integrate to 1, making it particularly well-suited for neural networks. Moreover, to our knowledge, there is a limited body of work applying EBM to time series analysis tasks.
>
>   ○ The model generates forecasts in one forward run unlike autoregressive models which suffers from inefficiency and error accumulation especially for long-term forecasts.
>
>   ○ The proposed versatile model demonstrates its effectiveness in various tasks, including short-term and long-term forecasting, imputation, pre-training, transfer learning (Appendix F), and uncertainty quantification (Appendix G). These extensive experiments validate the model's performance using the current model design.
>
>   ○ The model's pre-training capability makes TF-EBM a promising candidate for a time series foundation model to scale-up time series training.
>
> W2.Meanwhile, the author missed some pioneering literature about basic EBM
>
> ● We have added the pioneering literature mentioned about basic EBM in the updated paper.

---

> > ### Author Response · Authors · 2023-11-22
> > **Feedback Request**
> >
> > Dear reviewer F51B:
> >
> > With the author/reviewer discussions concluding in just a day, we would be grateful if you could confirm whether our response adequately addresses your primary concerns. If so, we respectfully request that you reconsider the score.
> >
> > Your further guidance on the paper and/or our rebuttal would be highly valued. We remain open to further discussion and paper refinement. To ensure ample time to address any lingering or new questions, we kindly request that our next round of communication be scheduled accordingly.
> >
> > Thank you sincerely for your dedication to enhancing our work on time series learning.

---

> > > ### Comment · Reviewer_F51B · 2023-12-04
> > >
> > > Dear Authors,
> > >
> > > Part of my concerns have been addressed. I agree that EBM is a rational choice comparing to autoregressive model in terms of error accumulation. The revised paper is easy to understand. Although I understand that the empirical results has proven the effectiveness of the proposed methods, I would encourage the author carefully investigate the roles of short-run dynamics. For example, what will happen if you sample a long run?
> > >
> > > I would raise my score to borderline accept.
> > >
> > > Best,

---

### Author Response · Authors · 2023-11-18
**General response to all reviewers**

We sincerely thank all the reviewers for their valuable feedback. We carefully revised our paper according to the commments of the reviewers and provide a detailed response to each reviewer. Here we list the major revision of the updated paper and hope our response will address your concerns.

● In response to the feedback from Reviewer F51B, we have added the reference of pioneering literature about basic Energy-Based-Models.

● In response to the feedback from Reviewer 5STj/DVHZ, we have made several updates to the paper. Firstly, Figure 1 and its corresponding caption, which illustrate the model architecture, have been revised. Additionally, equation (5) and (6) have been modified, along with the descriptions in Section 3.5 pertaining to the "encoder architecture and time-frequency module". These changes aim to enhance the clarity of the model framework. Furthermore, we have reorganized certain tables and figures within the paper, ensuring a more reader-friendly layout for improved comprehension.

● In response to uncertainty estimation concern from Reviewer op54/yqBZ, We add Appendix G of the updated paper dedicated to uncertainty quantification experiments.

● In response to transfer leanring concern from Reviewer yqBZ, we evaluated our model's transfer learning capability by pre-training it on the electricity dataset and fine-tuning it on the traffic and four ETT datasets. The results are summarized in Table 9 in Appendix F.

● In response to Time Grad comparison concern from Reviewer yqBZ, we replicated the experiments from Table 1 using TimeGrad and evaluated the models based on CRPS measures. Additionally, due to space constraints, we included the results of long-term probabilistic forecasting in Table 8 in Appendix E . Note that we modified the format of Table 1 to enhance its compactness for the main paper.

● In response to long-term dependence concern from Reviewer yqBZ, we added additional ablation studies for long-term forecasts in Figure 4 in Appendix C.3.

● In response to long-term CRPS comparison concern from Reviewer 5STj, we included CRPS results for the long-term (forecast horizon equals 336 and 720) forecasting in Table 8 in Appendix E.

● In response to ablation study for full datasets concern from Reviewer 5STj, we performed the same ablation study on the exchange rate and traffic datasets as well. The results are included in Table 6 in the Appendix C.2, which also includes the average of the lag-1 autocorrelation, standard deviation, and four dominant power spectrums.

---

### Meta-Review · Area_Chair_qPvy · 2023-12-05

**Metareview:**

This paper explores energy-based modeling for time series data, introducing a novel encoder structure to parameterize the energy function in an energy-based model for long-term probabilistic forecasting and imputation. Experimental results showcase competitive performance in both forecasting and imputation tasks across diverse public datasets. The paper is well-written, well-organized, and easy to follow, with promising presented results. While the paper proposes a new architecture for the energy function, its overall novelty is deemed limited since the energy-based learning framework and the study of Energy-Based Models for sequence generation have been previously investigated. The rebuttal effectively addresses most raised concerns, but the absence of strong endorsement from reviewers during internal discussions categorizes the paper as borderline. Considering the marginal contribution and lack of unanimous strong support for acceptance, the Area Chair recommends rejecting the paper. The AC encourages the authors to incorporate the valuable suggestions provided by the reviewers in their revision for consideration at the next venue.

**Justification For Why Not Higher Score:**

This is a borderline paper. There is a lack of enthusiasm during discussions, and no reviewer strongly supports the paper. In light of these considerations, the Area Chair recommends rejecting the borderline paper.

**Justification For Why Not Lower Score:**

NA

---

### Decision · Program_Chairs · 2024-01-16

Reject